# Loss of NEDD8 in cancer cells causes vulnerability to immune checkpoint blockade in triple-negative breast cancer

Irineos Papakyriacou[1], Ginte Kutkaite [2,3], Marta Rúbies Bedós [1], Divya Nagarajan[1], Liam P. Alford[1], Michael P. Menden [2,4] & Yumeng Mao [1] ✉

Immune checkpoint blockade therapy aims to activate the immune system to eliminate cancer cells. However, clinical benefits are only recorded in a subset of patients. Here, we leverage genome-wide CRISPR/Cas9 screens in a Tumor-Immune co-Culture System focusing on triple-negative breast cancer (TNBC). We reveal that *NEDD8* loss in cancer cells causes a vulnerability to nivolumab (anti-PD-1). Genetic deletion of *NEDD8* only delays cell division initially but cell proliferation is unaffected after recovery. Since the *NEDD8* gene is commonly essential, we validate this observation with additional CRISPR screens and uncover enhanced immunogenicity in *NEDD8* deficient cells using proteomics. In female immunocompetent mice, PD-1 blockade lacks efficacy against established EO771 breast cancer tumors. In contrast, we observe tumor regression mediated by CD8+ T cells against *Nedd8* deficient EO771 tumors after PD-1 blockade. In essence, we provide evidence that *NEDD8* is conditionally essential in TNBC and presents as a synergistic drug target for PD-1/L1 blockade therapy.

Triple-negative breast cancer (TNBC) accounts for 10–15% of all breast cancer cases and it is the most aggressive and invasive breast cancer type with limited treatment options[1]. TNBC is characterized by the lack of estrogen and progesterone receptors, and shows no over-expression or amplification of human epidermal growth factor receptor 2[1]. Chemotherapy is the standard-of-care therapy for TNBC but patients with advanced disease often develop resistance and show poor clinical outcome[2]. Therefore, TNBC represents a significant unmet clinical need requiring new treatment options to bring benefits to the patients.

Reinvigoration of anti-tumor immunity through immune checkpoint blockade (ICB) therapy against the PD-1/L1 axis has generated unprecedented clinical responses in several cancer types and is currently one of the most extensively evaluated research areas in oncology[3]. The therapeutic potential of ICB therapy has been tested in multiple randomized, placebo-controlled phase 3 clinical trials in TNBC

patients with advanced disease. For example, pembrolizumab as a monotherapy did not outperform chemotherapy in a phase 3 clinical trial (KEYNOTE-119)[4]. In addition, combination of atezolizumab and nab-paclitaxel chemotherapy significantly prolonged the progression-free survival (PFS) in advanced TNBC patients (IMpassion130)[5]. However, benefits on overall survival (OS) did not reach statistical significance[6], nor was validated in a confirmatory trial, i.e., IMpassion131[7]. Combining pembrolizumab with chemotherapy significantly improved PFS and OS of advanced TNBC patients and has been approved by the FDA, if stratified for PD-L1 positive tumors[8]. These results demonstrate the potential of immunochemotherapy but also highlight the clinical challenges in TNBC, including disease heterogeneity, choice of chemotherapy, genetic background of cancer cells, as well as the lack of validated biomarkers for patient stratification[9].

In order to map the immune-regulatory landscape in cancer cells, genome-wide CRISPR/Cas9 screens have been employed in co-cultures

[1]Science for Life Laboratory, Department of Immunology, Genetics and Pathology, Uppsala University, Uppsala, Sweden. [2]Computational Health Center, Helmholtz Munich, Neuherberg, Germany. [3]Department of Biology, Ludwig-Maximilians University Munich, Martinsried, Germany. [4]Department of Biochemistry and Pharmacology, University of Melbourne, Parkville, VIC, Australia. ✉e-mail: Yumeng.Mao@igp.uu.se

of genetically engineered human cytotoxic T cells and human cancer cells. Essential genes for efficient killing of human melanoma cells by T cell receptor (TCR)-transduced T cells have been identified and validated[10]. When co-cultured with chimeric antigen receptor (CAR)-modified T cells, defects in the death receptor pathways enabled leukemic cell survival and escape of T cell-mediated killing[11]. A recent study also employed genome-wide CRISPR activation screens to identify melanoma cancer intrinsic resistance to genetically modified human T cells[12]. These previous studies reveal deep mechanistic insight on the recognition of human cancer cells by T cells and could have a significant impact on the clinical implementation of adoptive cell therapy.

In this study, we aim to reveal and validate cancer vulnerabilities to ICB drugs in human TNBC cells. This is achieved by performing genome-wide CRISPR/Cas9 screens in a Tumor-Immune co-Culture System (TICS) that has been designed to investigate clinically approved ICB antibodies[13]. We identify that gRNAs targeting the *NEDD8* gene are significantly depleted from TNBC cells in the presence of nivolumab, suggesting its role as a TNBC vulnerability to ICB treatment. Further mechanistic investigations using advanced human cell assays and syngeneic mouse models confirm the strong immunogenic effects and anti-tumor efficacy as a result of *Nedd8* deletion in ICB-treated TNBC cells. In addition, our data reveal that essentiality of some "common essential" genes, such as *NEDD8*, can be compensated during cell reprogramming. We propose that targeting protein neddylation could enhance response to ICB drugs in TNBC patients. However, current pharmacological inhibitors against protein neddylation should be optimized due to the inhibitory effects on immune cells and potential off-target liabilities.

## Results

### Genetic screens identify *NEDD8* as a cancer vulnerability to ICB in human TNBC

To perform mechanistic investigation of clinically approved ICB drugs, i.e., nivolumab and durvalumab, we optimized a human Tumor-Immune co-Culture System (TICS), where primary human lymphocytes from healthy blood donors were co-cultured with human cancer cells. As shown in Fig. 1a, a human TNBC cell line, MDA-MB-231, significantly enhanced the release of granzyme B and interferon γ (IFNγ) in the presence of nivolumab or durvalumab in a ratio-dependent manner. To prove that the activation of primary human lymphocytes in TICS was dependent on antigens presented by TNBC cancer cells, we interrupted antigen presentation to CD8+ T cells by either genetic deletion of the *B2M* gene in cancer cells or by using a blocking antibody against HLA-ABC (Supplementary Fig. 1a). This abolished the proliferation of CD8+ T cells, but had no effects on the proliferation of CD4+ T cells and natural killer (NK) cells primed by cancer cells in the same experiment (Supplementary Fig. 1b).

After several optimization steps, TICS was adapted to enable genome-wide CRISPR screens to reveal genes that conferred vulnerability to ICB drugs in MDA-MB-231 cells (Fig. 1b). In brief, Cas9+ human TNBC cell line MDA-MB-231 (Supplementary Fig. 1c, d) was transduced with the Brunello gRNA library at an optimized MOI, according to an established protocol[14]. Library-transduced cells were cultured for 10 days to allow gene deletion and then co-cultured with freshly isolated primary lymphocytes from a healthy blood donor, ±10 μg/ml nivolumab. At the end of the co-culture on day 6, lymphocytes were gently washed away and cancer cells were harvested. Of note, we observed clear differences in medium consumption and total number of alive cancer cells when nivolumab was added, due to enhanced lymphocyte activation (Supplementary Fig. 1e). Frequencies of gRNAs were quantified in cancer cells using next generation sequencing and ranked according to the essentiality scores using the MAGeCK pipeline[15]. To increase the robustness of results, two independent screens were performed using lymphocytes from different donors at low or high lymphocyte-to-cancer (L2C) ratio (Fig. 1b and Supplementary Data 1).

To reveal genes controlling immune-mediated TNBC killing without nivolumab, we compared enriched and depleted gRNAs between co-culture and cancer cells cultured alone. Among the enriched genes, we identified known hits that are important for immune-mediated cancer killing, e.g., *STAT1* and *IFNGR2* (Supplementary Fig. 2a, b). In contrast, immune inhibitory genes in cancer cells, e.g., *ENPP1, CTNNB1, PRMT5*, were depleted after lymphocyte-cancer co-culture (Supplementary Fig. 2a, b).

Next, we sought to identify candidate genes that represent cancer vulnerability to PD-1 blockade therapy. Top depleted gRNAs in both screens were selected according to the distribution of essentiality scores using a cut-off of mean minus 2 standard deviations (SD) (Supplementary Fig. 2c). This resulted in 9 commonly depleted genes (*NEDD8, EIF2S2, NPIPB9, FAM86B2, PTDSS1, CRNKL1, SLC38A6, FOXG1* and *ZC3HAV1*), when comparing nivolumab-treated co-culture and co-culture alone (Fig. 1c). Because the *NEDD8* gene was strongly depleted (Supplementary Fig. 2d, e) and all 4 gRNAs targeting the *NEDD8* gene showed robust performance (Fig. 1d), we propose that it confers resistance to ICB therapy.

In order to explore the association between *NEDD8* mRNA expression and response to ICB therapy in breast cancer patients, we explored published RNA sequencing results from the I-SPY2 neoadjuvant platform trial (NCT01042379)[16], where patients received paclitaxel or paclitaxel in combination with pembrolizumab. In the chemo-immunotherapy arm, 44.9% of patients ($n = 69$) experienced a pathologic complete response (pCR). When stratified by *NEDD8* mRNA expression, we identified worse response in *NEDD8* high patients (22.2%), as compared to *NEDD8* medium (51.9%) or low (45.5%) subgroups (Fig. 1e).

### Loss of *NEDD8* is compensated by alternative pathways in human TNBC cells

NEDD8 protein is required for post-translational modification through protein neddylation[17]. To study its function in human TNBC cells, we deleted the *NEDD8* gene in three human TNBC cell lines, i.e., MDA-MB-231, HCC1937 and BT549, by transfecting RNP complexes containing a *NEDD8* targeting gRNA, i.e., crRNA+tracrRNA. Control cells were generated at the same time by transfecting RNP complexes without the *NEDD8*-targeting crRNA (Fig. 2a). Consistent with the public knowledge[18–25] of *NEDD8* being a common essential gene in >1000 human cancer cell lines (https://depmap.org/portal/achilles/), we observed a substantial decrease of cell viability after transfection of the *NEDD8*-targeting gRNA. To our surprise, *NEDD8* deficient cells recovered with time and proliferated at the same rate as the control cells (Fig. 2b).

Because it was important to confirm that the loss of NEDD8 protein expression translated to gene essentiality, we performed genome-wide loss-of-function CRISPR screens in the MDA-MB-231 wild-type (WT) and *NEDD8* knock-out (KO) cell line pair. Counts of gRNAs were compared between day 21 and day 4 after library introduction (Supplementary Data 1). Our results in MDA-MB-231 WT cells showed a strong agreement to a gene essentiality screen obtained from DepMap (Supplementary Fig. 3a) and the *NEDD8* gene was among the top-ranked essential genes in both screens (Fig. 2c and Supplementary Fig. 3b). In contrast, gRNAs targeting the *NEDD8* gene were not significantly changed in the KO cells between the 2 time points (Fig. 2c), confirming cell line recovery after *NEDD8* loss.

We identified uniquely essential genes in the WT cells, e.g., *GPX4*, or in the KO cells, e.g., *UBE2M, MTOR, RPTOR, RHEB* (Fig. 2c). In accordance, a pharmacological inhibitor against GPX4, i.e., ML210, preferentially inhibited proliferation of WT cells but was not effective on *NEDD8* KO cells (Fig. 2d), despite comparable GPX4 protein expression levels in the cell line pair (Supplementary Fig. 3c).

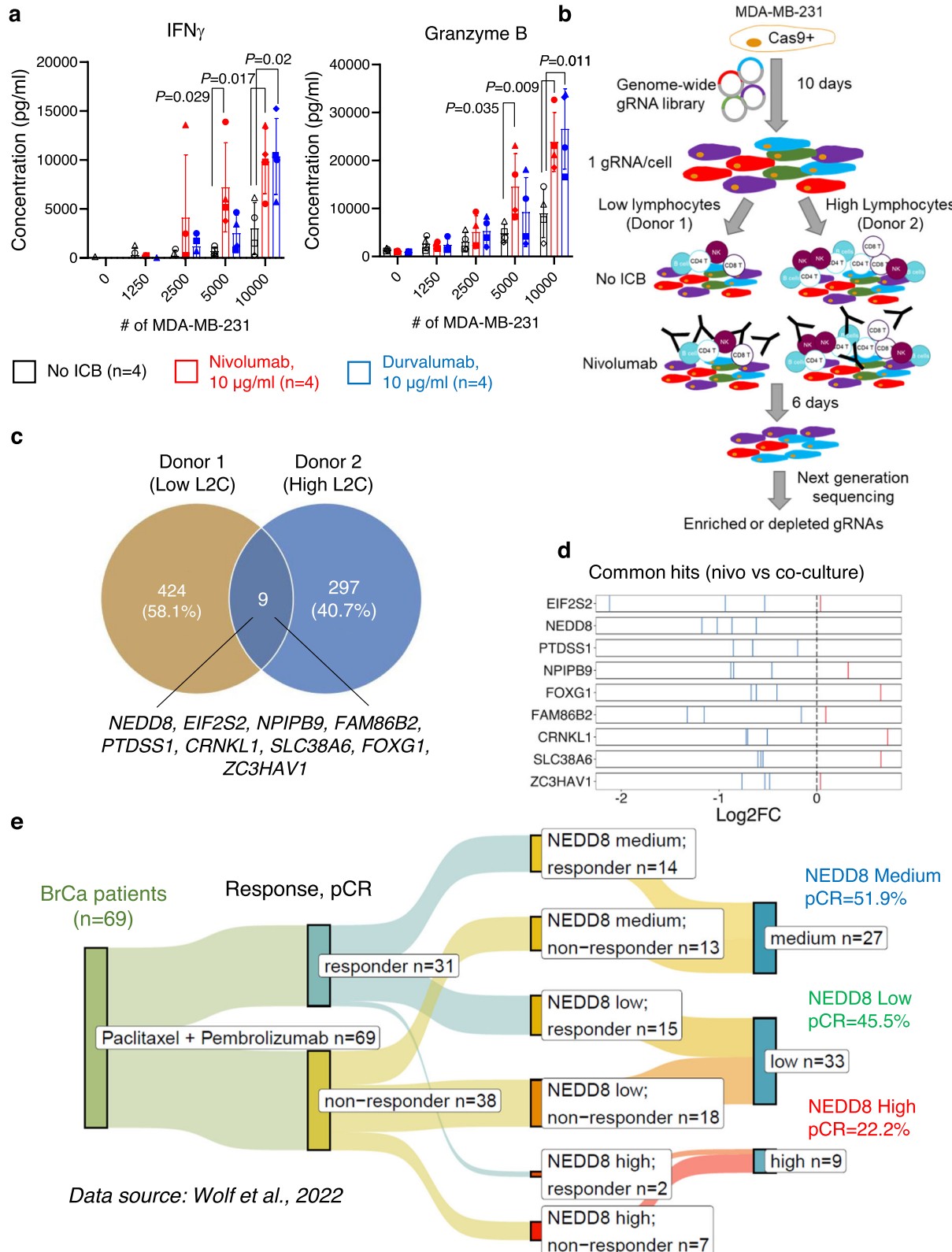

**Fig. 1 | Identification of *NEDD8* as a TNBC vulnerability against nivolumab in genome-wide CRISPR screens. a** Primary human lymphocytes (300,000 per well) were co-cultured with MDA-MB-231 cells in a 96-well flat bottom plate ±10 μg/ml nivolumab (red) or durvalumab (blue). Levels of soluble granzyme B or interferon γ (IFNγ) in culture supernatants were measured on day 5 by ELISA, mean ± SD, unpaired two-tailed *T*-test. Each symbol represents an individual lymphocyte donor (*n* = 4). **b** Schematic illustration and (**c**) the 9 overlapping hits from the genome-wide CRISPR screens when comparing co-cultures ±10 μg/ml nivolumab. **d** Demonstration of the 9 commonly depleted genes according to individual gRNAs performance (depleted gRNAs in blue and enriched gRNAs in red). **e** Analysis of the clinical relevance of *NEDD8* mRNA expression in breast cancer patients receiving paclitaxel in combination with pembrolizumab (*n* = 69) as part of the I-SPY2 neoadjuvant platform trial. Source data are provided as a source data file.

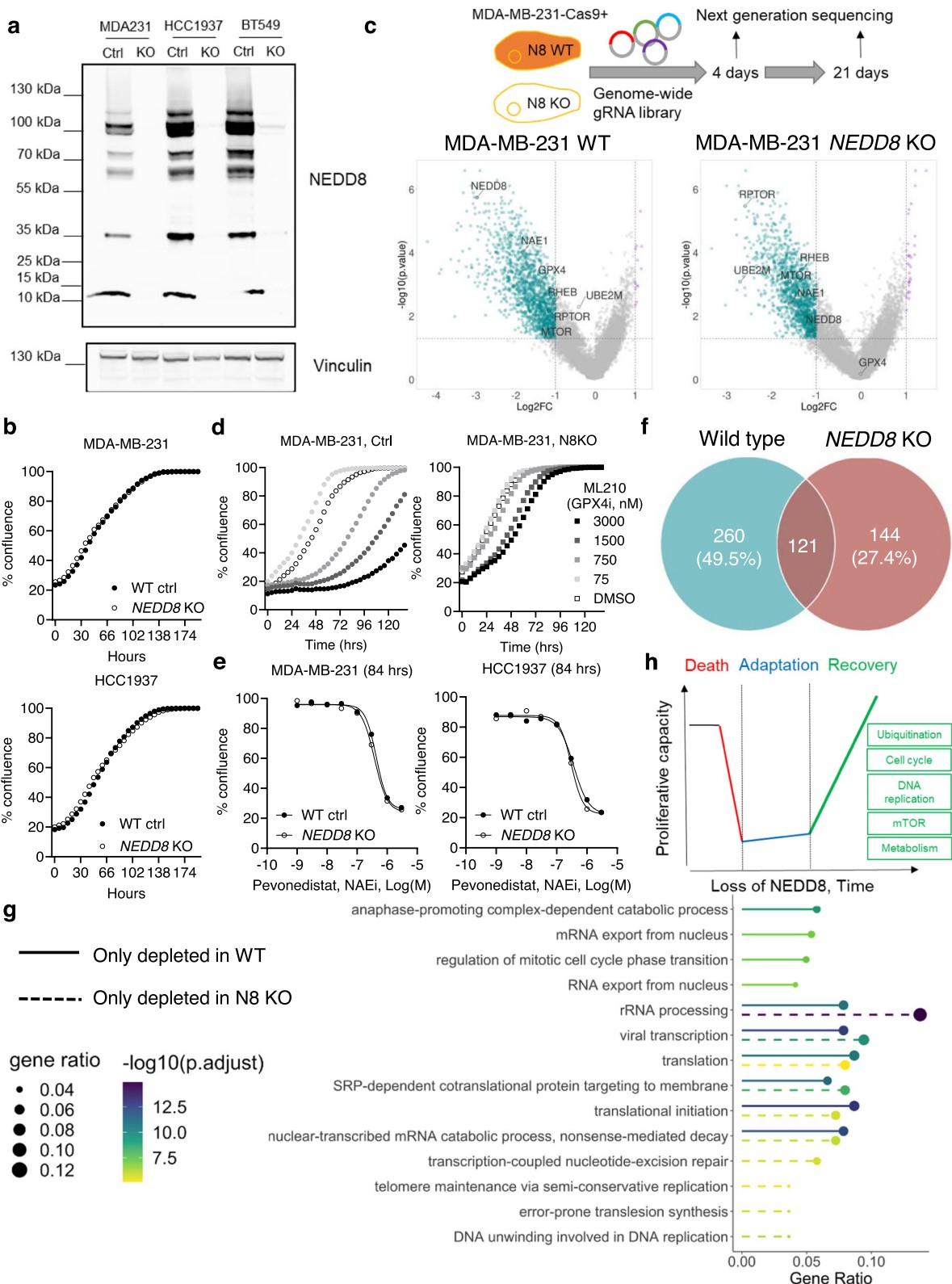

Moreover, we observed that gRNAs targeting the *NAE1* gene were depleted in both the WT and KO cells in the genome-wide screens (Fig. 2c). *NAE1* encodes the NEDD8-activating enzyme E1 subunit 1 (NAE1), which is a key subunit of the first heterodimer enzyme of the neddylation pathway[17]. Pharmacological inhibitors against NAE, i.e., pevonedistat[26] and TAS4464[27], have been developed and tested in patients as potential anti-cancer therapies[26,28]. Cell proliferation assays showed that the WT and KO cells were equally sensitive to

pevonedistat (Fig. 2e) and TAS4464 (Supplementary Fig. 3d). Deletion of *NEDD8* did not influence the expression of NAE1 protein (Supplementary Fig. 3e).

In order to map pathway changes in the WT/KO cell line pair, we selected strongly depleted gRNAs according to the distribution of essentiality scores using a cut-off of mean minus 3 SD (Supplementary Fig. 3f). This resulted in depleted genes unique to the WT cells ($n = 260$) and the KO cells ($n = 144$), as well as 121 genes that were

**Fig. 2 | Rescue of cell proliferation by alternative mechanisms in *NEDD8* deficient cells.** The *NEDD8* gene was deleted using CRISPR/Cas9 in three human triple-negative breast cancer (TNBC) cell lines, i.e., MDA-MB-231, HCC1937 and BT549. **a** Expression of the NEDD8 protein was measured using Western Blotting (representative blot of three independent experiments was shown) and (**b**) cell proliferation of the wild-type control (WT ctrl) and *NEDD8* knock-out (KO) cells was quantified in a live-cell imaging system. Representative experiment of three independent experiments were shown. **c** Genome-wide CRISPR screens were performed in MDA-MB-231 WT or *NEDD8* KO cells and the gRNA frequencies were compared between day 21 and day 4 (depleted genes in blue and enriched genes in purple). Data were processed in the MAGeCK pipeline and *p* values were calculated from the negative binomial model. Log2 fold changes were plotted against the Log10 *p* values in volcano plots with highlighted gene hits. One genome-wide CRISPR screen

was performed. **d** MDA-MB-231 WT or *NEDD8* KO cells were treated with the glutathione peroxidase 4 (GPX4) inhibitor (ML210), and the dose-dependent effects on cell proliferation were quantified using a live-cell imaging system. Representative experiment of three independent replicates. **e** Potency of a (NEDD8-activating enzyme) NAE inhibitor, pevonedistat, on the WT or *NEDD8* KO TNBC cell lines was shown at 84 h. Representative experiment of three independent replicates. **f** Number of uniquely or commonly depleted genes in the WT MDA-MB-231 or *NEDD8* KO cells was shown in a Venn diagram. **g** Pathway analysis on uniquely depleted genes in MDA-MB-231 WT or *NEDD8* KO cells in the genome-wide CRISPR screens. Enrichment analysis was conducted using hypergeometric test and Benjamini–Hochberg adjusted *p* values are reported. **h** Illustration of the conditional essentiality model of the *NEDD8* gene in TNBC cells. Source data are provided as a source data file.

depleted in both cell lines (Fig. 2f). Using the over-representation analysis, we revealed biological processes that became important upon *NEDD8* deletion, e.g., DNA replication (Fig. 2g). However, *NEDD8* deficient cells did not show enhanced sensitivity to chemotherapeutic drugs, e.g., paclitaxel, doxorubicin or fludarabin (Supplementary Fig. 4a). In contrast to the WT cells, *NEDD8* KO cells appeared to rely on distinct genes to sustain key cellular processes including translation and rRNA processing (Fig. 2g). This led us to a model, where the essentiality of certain "common essential" genes, e.g., *NEDD8*, is conditional due to system redundancy and cell proliferation can be rescued by alternative mechanisms, e.g., ubiquitination (Fig. 2h).

## NEDD8 controls global protein expression in human TNBC cells

Because protein neddylation is a key post-translational modification mechanism, we hypothesized that *NEDD8* deficiency can modulate global protein expression in TNBC cells. To test this hypothesis, we performed label-free protein quantification using mass spectrometry (Supplementary Data 2). Importantly, NEDD8 protein was detected only in the WT cells but not in the KO cells, validating the robustness of protein deletion as well as our previous results. With a cut-off threshold of FDR < 0.2 and an absolute Log2FC > 0.4, we identified 57 upregulated and 64 downregulated proteins in NEDD8 deficient MDA-MB-231 cells, as compared to the WT controls (Fig. 3a).

Pathway analysis demonstrated that NEDD8 deletion led to upregulated proteins in several pathways, including DNA replication and metabolic process (Fig. 3b). An in-depth analysis of the protein interaction network demonstrated that proteins for cell cycle and DNA replication, as well as compound metabolism, were upregulated in NEDD8 KO cells. In contrast, NEDD8 deficient cells showed attenuated protein expression for epidermal cell differentiation, cytoskeleton and chromatin organization (Fig. 3c). These findings were in line with data from the genome-wide CRISPR screens (Fig. 2c, g), where genes regulating DNA replication and mTOR/metabolic pathway became more essential in the KO cells.

In particular, our analysis revealed reprogramming of the post-translational modification in the absence of NEDD8 (Fig. 3c). Multiple regulatory enzymes, e.g., UBE2T, UBE3C, UBE4A and SMURF2, increased in expression in NEDD8 KO cells, which could serve as compensatory mechanisms to sustain protein homeostasis and global ubiquitination (Supplementary Fig. 4b). Although UBE2T was not detected in wild type cells using proteomics, i.e., "unique in KO", we demonstrated a low expression using western blotting and confirmed its upregulation upon NEDD8 deletion (Fig. 3d).

To functionally validate whether protein ubiquitination became indispensable in KO cells, we tested the effects of an ubiquitin E1 enzyme (UBA1) inhibitor, i.e., TAK-243, on the proliferation of control or NEDD8 KO MDA-MB-231 cells. Indeed, TAK-243 more potently inhibited the proliferation of NEDD8 KO cells, as compared to the control cells (Fig. 3e). Of note, both pevonedistat and TAK-243 induced the stabilization of CDT1 in a dose-dependent manner (Fig. 3f, g),

which is a known cytotoxic mechanism in pevonedistat-treated cells[29]. While stabilization of CDT1 was comparable between KO and control cells treated with pevonedistat (Fig. 3f), TAK-243 induced a stronger effect in KO cells at low concentrations (Fig. 3g). Pevonedistat, but not TAK-243, strongly inhibited the modification of cullin-1 in control and KO MDA-MB-231 cells (Supplementary Fig. 4c).

## Deletion of NEDD8 in TNBC cells enhances immune activation driven by ICB drugs

Upon NEDD8 deletion, we identified enhanced protein expression for antigen presentation (HLA-DRA, -DRB and CD74) among immune regulatory proteins (Figs. 3a, c and 4a). Subsequent experiments performed in flow cytometry confirmed that NEDD8 deletion led to enhanced expression of HLA-DR on MDA-MB-231 and HCC1937 cell lines (Fig. 4b). Treatment of TNBC cells with IFNγ induced surface expression of HLA-DR, which was further enhanced in the absence of NEDD8 (Fig. 4b). Of note, NEDD8 deletion in TNBC cells demonstrated similar effects on the expression of HLA-DR as compared to treatment with IFNγ (Fig. 4b), indicating strongly enhanced immunogenicity in KO cells. The down-regulation of surface CD55 on NEDD8 deficient human TNBC cells was also validated by flow cytometry (Supplementary Fig. 4d). However, NEDD8 deficiency did not modulate surface expression of HLA-ABC, PD-L1, IFNγRα on human TNBC cell lines at the baseline or after IFNγ treatment (Supplementary Fig. 4e).

To test whether NEDD8 KO TNBC cells can induce stronger immune cell activation, control or NEDD8 deficient MDA-MB-231 cells were co-cultured with primary human lymphocytes in TICS ± ICB drugs. Induction of soluble IFNγ and granzyme B by cancer cells were observed in culture supernatants and KO cells induced a marginal enhancement, as compared to the control cells (Fig. 4c). In accordance with previous results (Fig. 1a), we observed significantly increased production of these immune-activating cytokines in the presence of nivolumab or durvalumab, which was further enhanced by NEDD8-deficient cells (Fig. 4c and Supplementary Fig. 5a). Similar results were observed when assessing the proliferation of CD8+ and CD4+ T cells in TICS in response to either nivolumab or durvalumab (Fig. 4d and Supplementary Fig. 5b). The activation of NK cells by NEDD8 KO TNBC cancer cells showed only a trend of increase (Supplementary Fig. 5c). The increased release of soluble granzyme B in response to nivolumab or durvalumab was confirmed using an additional control/KO cell line pair derived from HCC1937 cells (Fig. 4e).

To examine the mechanistic insights of NEDD8 in cancer immunogenicity, we re-expressed a truncated form of the NEDD8 protein in MDA-MB-231 KO cells, i.e., NEDD8-T. NEDD8-T lacked the C-terminus diglycine residues[30–32] and therefore failed to conjugate to enzymes or substrates (Fig. 4f). In TICS, NEDD8-T cells demonstrated equally potent induction of immune-activating cytokines in response to ICB drugs, as compared to NEDD8 KO cells (Fig. 4g). Of note, NEDD8-T cells remained sensitive to pevonedistat (Supplementary Fig. 5d).

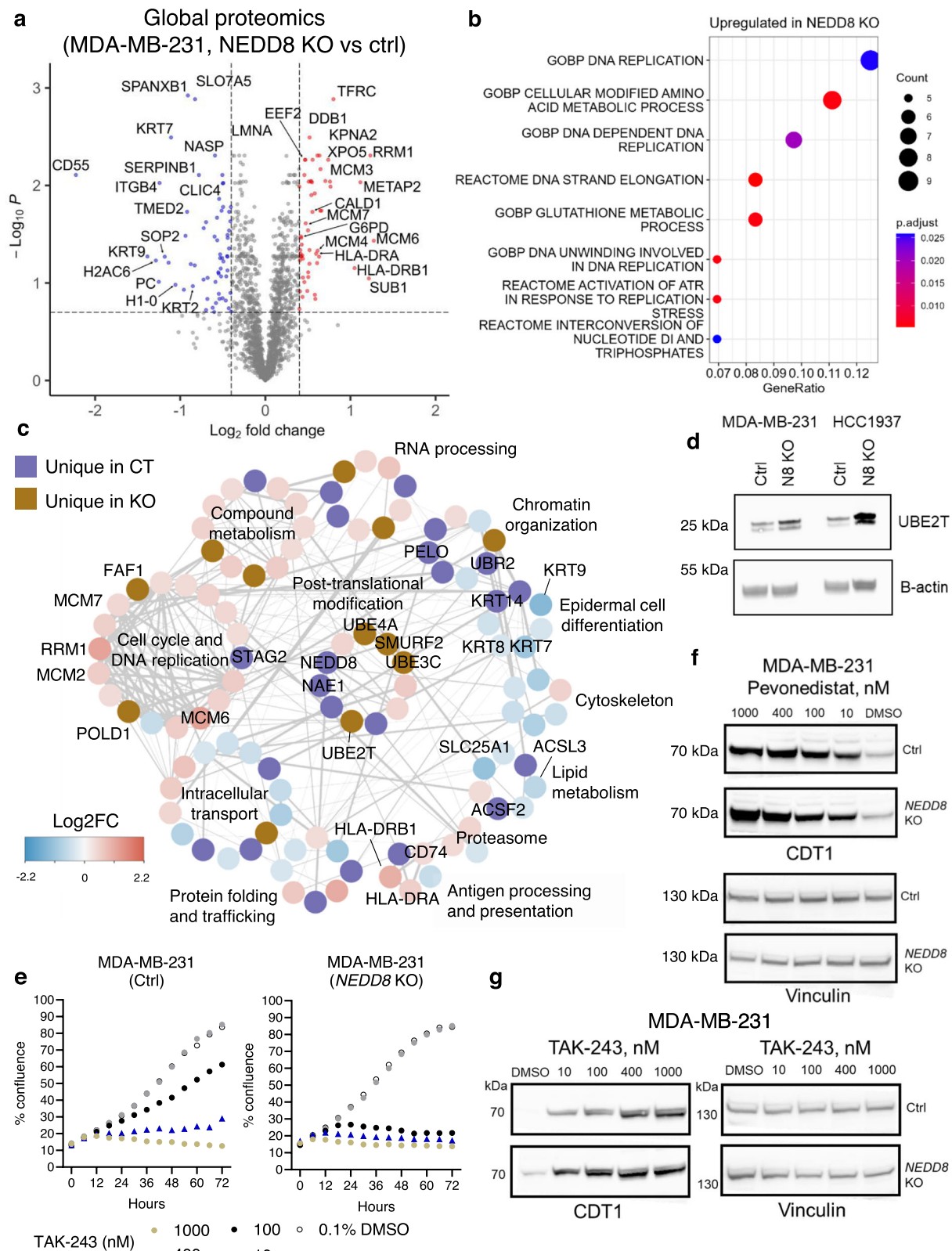

**Pharmacological inhibition of protein neddylation modulates cancer-driven immune activation**

Next, we sought to test whether inhibition of protein neddylation by NAE inhibitors can potentiate TNBC cancer-driven immune activation. As shown in Fig. 5a, pevonedistat led to a dose-dependent inhibition of protein neddylation in MDA-MB-231 cells without clear effects on the expression of free NEDD8 protein (~9 kDa). Monitored in real-time by Incucyte, pevonedistat inhibited the proliferation of 3 human TNBC cell lines in vitro with IC50 values between 180 nM and 600 nM (Supplementary Fig. 5e). When tested on primary human lymphocytes activated with αCD3/28 beads ± rhIL2, pevonedistat demonstrated a negative impact on the proliferation of CD4+ T cells, CD8+ T cells and NK cells with comparable potencies as observed in cancer cells, i.e., between 100 nM and 600 nM (Fig. 5b).

**Fig. 3 | Regulation of global protein expression by NEDD8 in human triple-negative breast cancer (TNBC) cells.** Label-free protein quantification was performed using mass spectrometry in the MDA-MB-231 wild-type (WT)/NEDD8 knock-out (KO) cell line pair using 4 replicate samples of each line. **a** Up-regulated (red) and down-regulated (blue) proteins upon *NEDD8* deletion were shown in a volcano plot. A Welch's unequal variances *T*-test was applied to determine differences in protein expression between control and KO cells. The False Discovery Rate was calculated to adjust the *p* values. **b** Differentially expressed proteins and unique proteins were divided into either upregulated in NEDD8 KO or upregulated in control cells for pathway analysis. A hypergeometric test was conducted to determine enriched pathways from the Reactome and Gene Ontology Biological Process collections. *p* values were adjusted with Benjamini–Hochberg correction.

**c** Interaction of changed proteins in the WT/KO cell line pair was grouped based on biological processes. Purple: unique in WT, Brown: unique in KO. Color is based on Log2 fold changes between KO and WT cells. **d** Expression of UBE2T was measured by Western Blotting. Representative image of 3 independent repeats was shown. **e** Control or KO MDA-MB-231 cells were treated with a UBA1 inhibitor, TAK-243, at 1000, 400, 100, 10 nM or 0.1% DMSO. Cell proliferation was measured by live-cell imaging. Representative experiment of 2 independent repeats. Control or KO MDA-MB-231 cells were treated with (**f**) pevonedistat or (**g**) TAK-243 at 1000, 400, 100, 10 nM or 0.1% DMSO. Cells were harvested at 24 h and the expression of CDT1 was measured using western blotting. Representative western blot of 2 independent repeats. Source data are provided as a source data file.

---

To rule out that the immune inhibitory property was specific to pevonedistat, we tested a more potent NAE inhibitor, TAS4464[27]. Similar to the data from pevonedistat, TAS4464 potently inhibited the proliferation of TNBC cells (Supplementary Fig. 5f) as well as primary human T cells (Supplementary Fig. 6a). This suggested that current NAE inhibitors under clinical testing carry negative effects on primary human lymphocytes. In TICS, pevonedistat at 100 nM significantly enhanced the release of granzyme B and IFNγ in combination with nivolumab or durvalumab (Fig. 5c). However, the synergistic effects diminished at 1000 nM, possibly due to its direct inhibition on immune cells in the co-culture (Fig. 5c).

To assess the long-term effects of NAE inhibition on protein neddylation and TNBC immunogenicity, we generated treatment resistant cell lines by chronic exposure of MDA-MB-231 cells to pevonedistat in vitro (Fig. 5d). Of note, compound resistant MDA-MB-231 cells demonstrated elevated protein neddylation levels (Supplementary Fig. 6b), which remained sensitive to pevonedistat (Supplementary Fig. 6c), ruling out treatment-driven pathway mutations[33,34]. Of note, resistant cells triggered significantly weaker release of IFNγ and granzyme B in response to ICB antibodies in TICS, as compared to the parental cell line (Fig. 5e).

To investigate whether pevonedistat-induced protein neddylation conferred immune resistance, we deleted the *NEDP1* gene using CRISPR/Cas9 in MDA-MBA-231 cells (Supplementary Fig. 7a). NEDP1 removes NEDD8 from protein substrates[30] and as expected, *NEDP1* KO cells demonstrated substantial accumulation of neddylated enzymes and substrates (Supplementary Fig. 7b). However, its deletion did not result in reduced immune activation in TICS (Supplementary Fig. 7c).

### Neddylation inhibitors target protein ubiquitination
Because pevonedistat was able to inhibit cells lacking NEDD8 protein (Fig. 2e) or functional protein neddylation (Supplementary Fig. 5d), we speculated that off-target mechanisms may contribute to the phenotype observed in drug-resistant cells. Using CRISPR/Cas9, we silenced the *NAE1* gene in MDA-MB-231 cells (Fig. 5f), which is the putative target for neddylation inhibitors. Pevonedistat efficiently inhibited the proliferation of *NAE1*-deficient cells (Fig. 5g), demonstrating compound mode-of-action that are unspecific to neddylation.

Because protein ubiquitination and neddylation are closely related, we sought to investigate whether neddylation inhibitors could affect ubiquitination. As shown in Fig. 5h, pevonedistat at 1000 nM clearly reduced the total ubiquitin levels in MDA-MB-231 cells, which coincided with the negative effects on immune activation in TICS at this concentration (Fig. 5c).

When measuring the expression of NAE1 protein (62.7 kDa) in human TNBC cells, we observed a second band at ~70 kDa, which did not differ between control or NEDD8 KO cells (Supplementary Fig. 3e) but was not detectable in NAE1 KO cells (Fig. 5f). Treatment with UBA1 inhibitors TAK-243 (Fig. 5i) or PYR41 (Supplementary Fig. 7d), as well as neddylation inhibitor pevonedistat (Fig. 5j) diminished the expression of this band.

Given the clear negative impact of NAE inhibitors on immune cells and cancer immunogenicity due to off-target effects, we decided to employ CRISPR/Cas9 to specifically target the *Nedd8* gene in murine cancer cells for in vivo studies.

### Genetic deletion of *Nedd8* in cancer cells enhances anti-tumor efficacy of PD-1 blockade
Because immune activation relied on allogeneic antigens presented by cancer cells in TICS, we decided to validate the cancer intrinsic role of the *Nedd8* gene using syngeneic mouse models. Expression of the *Nedd8* gene was disabled using CRISPR/Cas9 in a murine breast cancer cell line, EO771 (Fig. 6a). Similar to human TNBC cells, the proliferation of EO771 murine breast cancer cells in vitro was comparable in the control/KO cell line pair after recovery (Fig. 6a). Next, we implanted the control or *Nedd8* KO EO771 cells subcutaneously (s.c.) on female C57BL/6 mice. When tumors were palpable, mice were treated with an αPD-1 mAb or a Rat IgG2a isotype control intraperitoneally (i.p.) on day 5, 8 and 11 (Fig. 6b). In mice bearing WT tumors, we observed a moderate response to PD-1 blockade. In contrast, PD-1 blockade resulted in highly significant tumor growth delay ($p < 0.0001$) in all mice bearing *Nedd8* KO EO771 tumors (Fig. 6b).

To assess the anti-tumor efficacy of PD-1 blockade in large tumors, we initiated the treatment when average tumor volumes reached ~50 mm³. None of the mice bearing control EO771 tumors responded to PD-1 blockade (Fig. 6c) and *Nedd8* deficiency did not delay tumor growth when treated with the isotype control antibody, as compared to mice bearing control tumors (Supplementary Fig. 7e). Strikingly, *Nedd8* deletion in EO771 cells significantly delayed the progression of established tumors in response to anti-PD-1 treatment ($p < 0.0001$), resulting in a 40% complete response (Fig. 6c). When treated with PD-1 blockade, mice bearing *Nedd8* deficient tumors showed significantly prolonged survival, as compared to mice treated with the IgG control ($p < 0.01$, Fig. 6c).

Because established EO771 control tumors are unresponsive to PD-1 blockade, we sought to prove that the potent anti-tumor efficacy in *Nedd8* KO tumors after PD-1 blockade was a result of immune-mediated cytotoxicity. Mice bearing *Nedd8* KO EO771 tumors were treated with a CD8 depleting antibody or a Rat IgG2b isotype control, 2 days before αPD-1 therapy with a 3-day interval (Fig. 6d and Supplementary Fig. 7f). Consistent with earlier results, *Nedd8* deficiency significantly improved response to PD-1 blockade and survival of tumor-bearing mice, which was abrogated with the depletion of CD8+ T cells (Fig. 6d).

### *Nedd8* deficient breast tumors exhibit a favorable intra-tumoral immune landscape
In order to dissect immunological changes in *Nedd8*-deficient breast tumors, we analyzed intra-tumoral immune cell population and mRNA gene signatures using flow cytometry and the Nanostring technology, respectively. Because PD-1 blockade induced tumor regression in mice bearing *Nedd8*-deficient tumors, we harvested tumor tissues 2 days after the last antibody infusion before complete regressions occurred

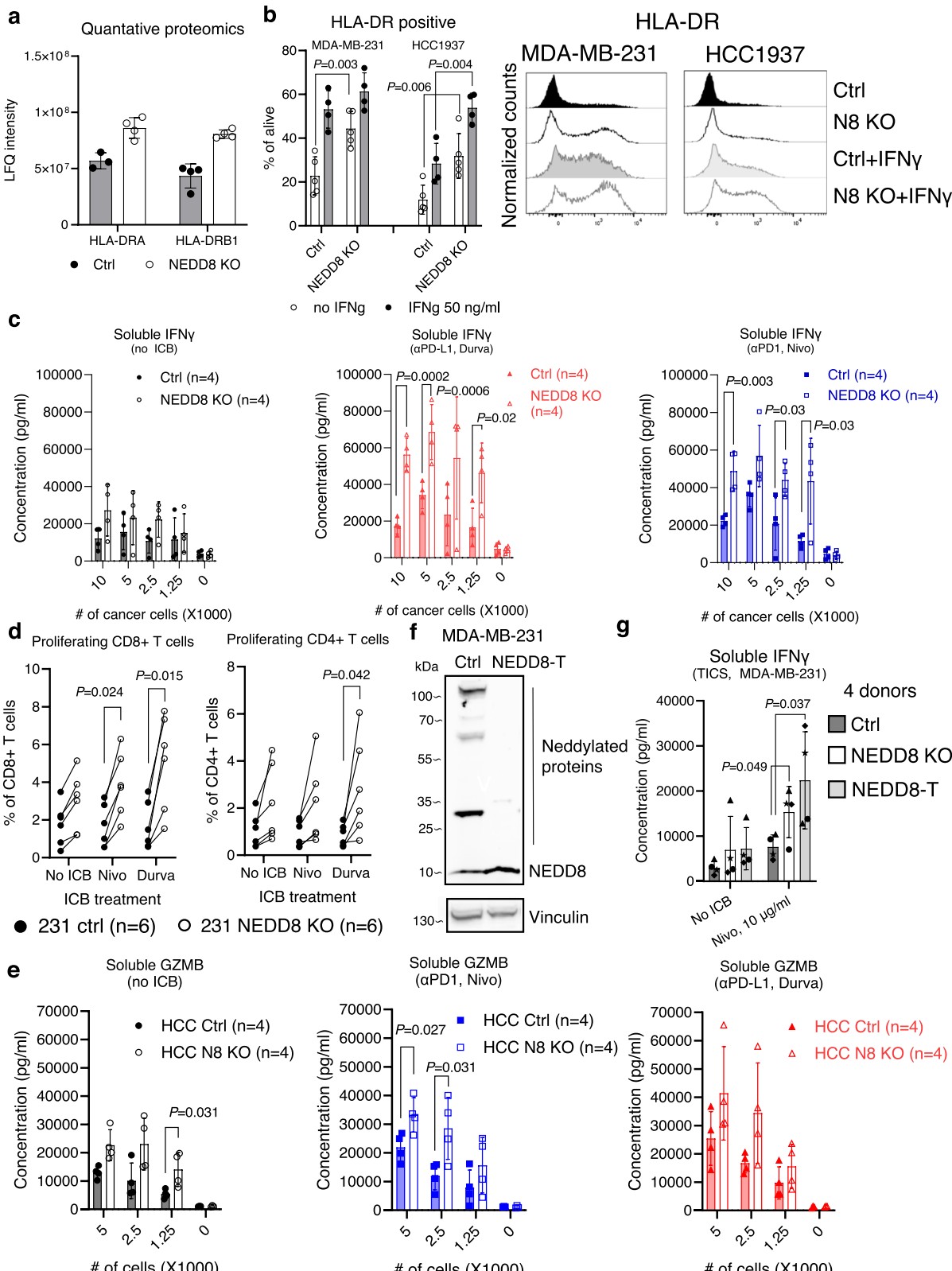

(Fig. 7a). At this study endpoint, PD-1 blockade was insufficient in controlling the growth of EO771 tumors but resulted in a non-significant delay in the growth of KO tumors (Fig. 7b).

Flow cytometric analysis using the gating strategy in Supplementary Fig. 8a revealed that frequencies of CD8+ T cells (Supplementary Fig. 8b) and regulatory T cells (Supplementary Fig. 8c), or the ratio between CD8+ and CD4+ T cells (Supplementary Fig. 8d) were

comparable among treatment groups. Notably, T cells in *Nedd8*-deficient tumors receiving ICB demonstrated a more functional phenotype with increased expression of surface CD25 (Fig. 7c) and intracellular tumor necrosis factor α (TNFα) (Fig. 7d), while surface expression of PD-1 remained comparable on T cells among groups (Supplementary Fig. 8e). Moreover, CD11b+ myeloid cells were reduced in KO tumors treated with PD-1 blockade, as compared to the

**Fig. 4 | Enhancement of immune activation by *NEDD8* deficient triple-negative breast cancer (TNBC) cells in response to immunotherapy drugs. a** Label-free quantification of peptides derived from HLA-DRA and HLA-DRB in MDA-MB-231 wild-type (WT) and NEDD8 knock-out (KO) cell lines in proteomics, 4 technical replicates. **b** WT or NEDD8 KO human TNBC cell lines, i.e., MDA-MB-231 and HCC1937, were treated with PBS (5 independent replicates) or 50 ng/ml rhIFNγ (4 independent replicates) for 24 h. Surface expression of HLA-DR was quantified using flow cytometry. WT or *NEDD8* KO MDA-MB-231 cells were co-cultured with CTV-pulsed primary human lymphocytes ±10 µg/ml nivolumab (red) or durvalumab (blue). **c** Release of soluble IFNγ was tested by ELISA (4 independent donors) or (**d**) proliferation of T cells was quantified by flow cytometry on day 5 (6

independent donors). **e** Ctrl or NEDD8 KO HCC1937 cells were co-cultured with primary human lymphocytes ±10 µg/ml nivolumab or durvalumab and release of soluble granzyme B was tested by ELISA on day 5 (4 independent donors). **f** A truncated NEDD8 protein lacking the C-terminus diglycine residues was re-expressed in NEDD8 KO MDA-MB-231 cells (NEDD8-T) and protein neddylation was measured using Western Blotting. Representative image of 2 independent repeats. **g** Control (dark gray), NEDD8 KO (open) or NEDD8-T (light gray) cells (2500 cells per well) were co-cultured with primary human lymphocytes ±10 µg/ml nivolumab and release of soluble IFNγ was tested using ELISA on day 5 (4 independent donors). All data in this figure were shown as mean ± SD and unpaired two-tailed *T*-test was used for statistical analysis. Source data are provided as a source data file.

IgG-treated control tumors (Fig. 7e). Among myeloid cells, we observed elevated number of activated macrophages in KO tumors treated with PD-1 blockade as compared to the KO tumors treated with the isotype control antibody (Fig. 7f), while Ly6G+ neutrophils showed a trend of reduction (Supplementary Fig. 8f).

To gain a broader view of the intra-tumoral immunological changes, we quantified the expression of immune-related genes using a Nanostring panel (Supplementary Data 3). Because PD-1 blockade was inefficient in EO771 tumors, only few genes changed upon therapy (Supplementary Fig. 8g). In contrast, *Nedd8* deficiency alone led to significant changes in immune-related pathways. Expression of genes associated with interferon response (*Mx1*, *Stat1*, *Ifitm1*, *Cxcl10*) and immune cell effector function (*Il2*, *Gzma*, *Tnfrsf8*) were significantly increased in KO tumors, as compared to control EO771 tumors (Fig. 7g). In line with the results from flow cytometry, *Nedd8*-deficient tumors presented less abundant mRNA transcripts, e.g., *Sirpa*, *S100a8*, *Csf1*, *Mmp9*, *Mmp12*, *Tgfbr1*, for myeloid cells with a suppressive phenotype (Fig. 7g). Addition of PD-1 blockade to *Nedd8*-deficient tumors sustained these immunological changes and potentiated antigen presentation, e.g., *Cd80*, and T cell activation, e.g., *Il2ra/Cd25* (Fig. 7h), which confirmed our earlier results using flow cytometry (Fig. 7c).

## Discussion

Triple-negative breast cancer (TNBC) is a heterogeneous disease and presents an immunosuppressive intra-tumoral landscape. Although TNBC cells show PD-L1 positivity[35] and the infiltration of T cells correlates to patient survival[36,37], PD-1 blockade therapy alone is yet to show clinical benefits in patients with advanced disease[4]. Moreover, clinical responses to immunotherapy in TNBC patients may be limited by additional factors, e.g., the immunosuppressive micro-environment in TNBC tumors, as well as the aggressive growth behavior and intrinsic resistant mechanisms of cancer cells[38,39]. Therefore, we hypothesize that key cancer vulnerability genes can be targeted to improve response to immunotherapy in TNBC patients.

Genome-wide loss-of-function or activation screening using CRISPR/Cas9 offers a powerful tool to uncover genes that are essential for cancer cell survival and response to therapy. Several studies have been performed in human cancer cells to reveal genes controlling cytotoxicity mediated by genetically engineered T cells[10–12]. Previously, identification of cancer vulnerability to ICB antibodies using CRISPR/Cas9 loss-of-function screens has been conducted in immunocompetent mouse models bearing syngeneic tumors[40–42]. In particular, the discovery of *Ptpn2* as a resistance gene to immunotherapy[40] has led to the development of a small molecule compound suitable for testing in patients[43]. While these studies are highly relevant, murine cancer cells resembling human TNBC have not been included.

Inspired by a study where healthy donor-derived T cells contain clones that recognize mutated cancer neoantigens[44], we have optimized a human Tumor-Immune co-Culture System (TICS) to investigate cancer-driven immune activation in response to ICB drugs[13]. Instead of using isolated CD8+ T cells, TICS utilizes unsorted human

lymphocytes in order to identify effective orthogonal cancer killing mechanisms mediated by HLA class II epitopes or NK cells[45].

Our genome-wide screens in TICS reveal that the *NEDD8* gene plays a crucial role in TNBC vulnerability against nivolumab. NEDD8 is a ubiquitin-like protein that governs protein neddylation, which is an important post-translational machinery[17]. Multiple earlier genetic screens unanimously demonstrated the essentiality of *NEDD8* in cell survival[18–24] and therefore *NEDD8* is regarded as one of the 'common essential' genes (or 'pan-essential' genes[20]).

Paradoxically, we observe and validate that TNBC cells recover from genetic targeting of *NEDD8* and proliferate at a comparable rate as the *NEDD8*-competent control cells. Combining proteomics and genome-wide CRISPR screens, we delineate the compensatory roadmap in TNBC cells upon *NEDD8* loss. Ubiquitination enzymes, DNA replication machinery and the mTOR pathway become important in maintaining cell proliferation in *NEDD8* deficient cells. It has been reported that atypical neddylation occurs through ubiquitin enzymes as a result of an increased NEDD8/ubiquitin ratio under stress condition[30–32,46]. However, it remains to be tested whether the absence of NEDD8 could lead to compensatory effects by the ubiquitin system. We show that global ubiquitination is not impaired in NEDD8 deficient cells and these cells become more sensitive to UBA1 inhibition. Therefore, it can be speculated that the loss of NEDD8 triggers cellular reprogramming and the ubiquitination system becomes indispensable in cancer cells.

Although exemplified with one gene in human cancer cells, our data highlight an opportunity to refine the common essentiality theory. Whilst the essentiality of many pan-essential genes is "absolute" to cancer cells, we propose that a subset of genes is "conditionally" essential. Loss of such genes triggers cellular reprogramming in cancer cells, which rescues cell survival through compensatory mechanisms. Further work is warranted to assess the validity of this concept in non-malignant cells.

Protein neddylation is frequently amplified in cancer cells to sustain cell proliferation and has been regarded as a promising target for anti-cancer therapy. Pharmacological inhibitors, i.e., pevonedistat and TAS4464, are designed to induce cancer cell death through disruption of the enzymatic function of NAE[26–29]. Motivated by the strong anti-proliferative effects on cancer cells, NAE inhibitors have been evaluated in patients[26–28]. However, pevonedistat failed to deliver clinical efficacy in patients with myeloid cell malignancies in the PANTHER phase 3 clinical trial[47]. In a phase 1 clinical study in patients with multiple myeloma, severe liver injury led to trial termination for TAS4464 (NCT02978235). Similar dose-limiting toxicity was observed for TAS4464 in patients with solid cancers in another clinical trial[48].

Our results confirm that human and murine breast cancer cells present active protein neddylation, which is abolished with NAE inhibitors in a dose-dependent manner. However, the strong anti-proliferative capacity of this class of chemical compounds is not exclusive to the inhibition of protein neddylation. This is because isogenic human TNBC cell lines lacking NEDD8/NAE1 protein or expressing a non-functional NEDD8 protein, remain sensitive to NAE

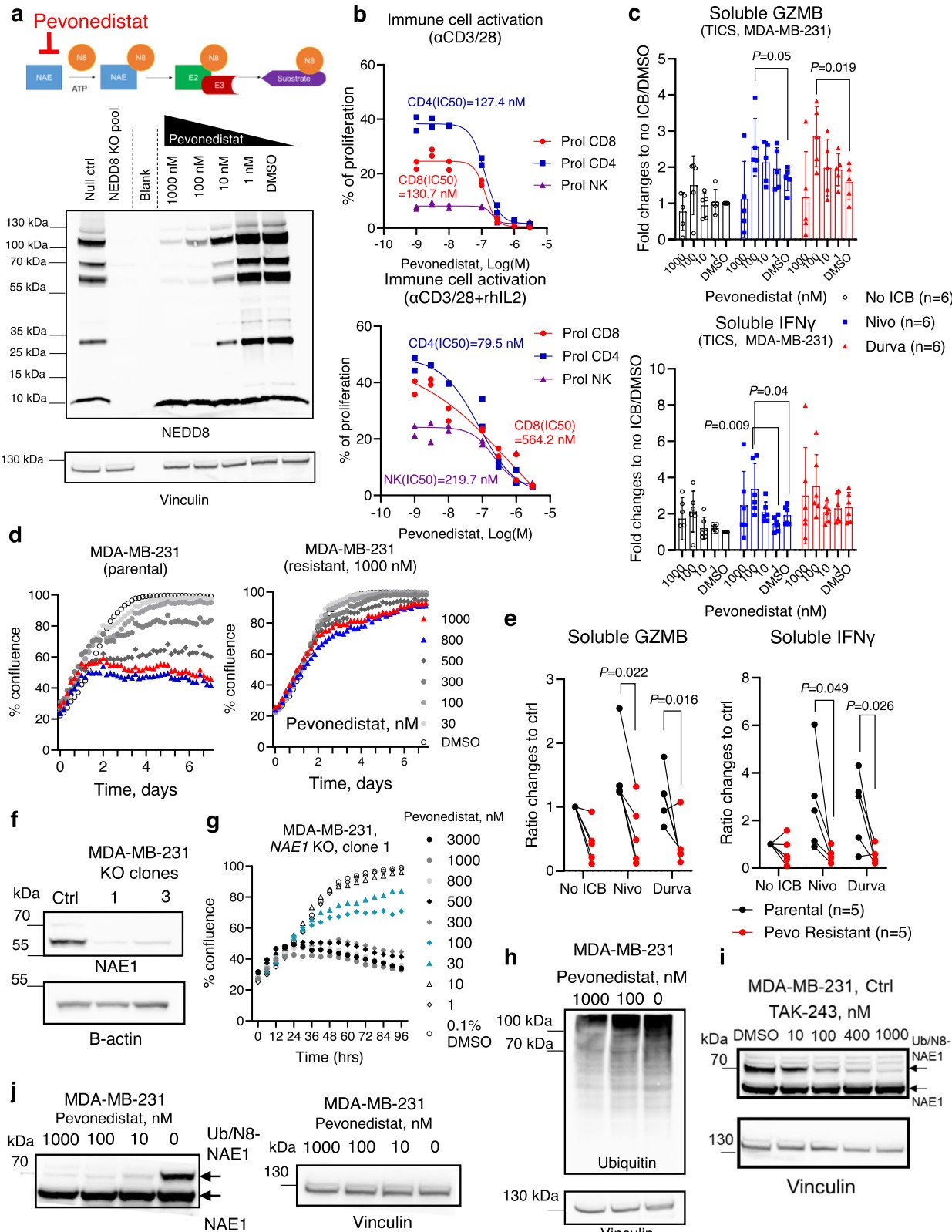

inhibitors. We further demonstrate that pevonedistat could dampen global protein ubiquitination at above IC50 concentrations, potentially due to the inhibition of cullin-RING ligases (CRLs)[29]. Because the modification of cullin-1 is detectable in KO cells, the identity of this modification remains to be tested.

In addition to the CRL-dependent mechanisms, neddylation of substrates could be conducted in a CRL-independent manner[17].

For example, mouse double minute 2 (MDM2) mediates neddylation of p53 and reduces its transcriptional activity through NAE1[49]. In accordance, CRL-independent neddylation blocks substrate ubiquitination, which can be reverted by pevonedistat treatment[50,51]. These findings are in line with our observation, where NEDD8 KO human TNBC cells show greater sensitivity to UBA1 inhibition, possibly due to increased dependency on the ubiquitination system.

**Fig. 5 | Modulation of cancer-driven immunity by neddylation inhibitors.**
**a** MDA-MB-231 cells were treated with pevonedistat for 24 h and the NEDD8 expression was assessed in Western Blotting. Representative image of 3 independent repeats was shown. **b** CellTrace Violet (CTV)-pulsed primary human lymphocytes were incubated with pevonedistat in the presence of αCD3/28 activation beads ±rhIL2 (100 ng/ml). The resulting cell proliferation was quantified by flow cytometry. Representative experiment from 3 independent donors was shown. **c** Primary human lymphocytes were co-cultured with MDA-MB-231 cells in the presence of pevonedistat ± nivolumab or durvalumab (10 µg/ml). Soluble granzyme B and IFNγ were quantified by ELISA on day 5. Six independent blood donors were included and data was shown at mean ± SD, unpaired two-tailed T-test. **d** Effect of pevonedistat on parental or pevonedistat resistant MDA-MB-231 cells was tested in a live-cell imaging system. Representative experiment of 3 independent repeats was shown. **e** Primary human lymphocytes were co-cultured with parental or pevonedistat-resistant MDA-MB-231 cells ± nivolumab or durvalumab

(10 µg/ml). Levels of soluble granzyme B and IFNγ were measured by ELISA on day 5. Five independent donors were included and data were shown with mean ± SD, unpaired two-tailed T-test. **f** NEDD8-activating enzyme 1 (NAE1) protein expression in control MDA-MB-231 cells or knock-out (KO) clones was measured using Western Blotting. Representative image of 2 independent repeats was shown. **g** The *NAE1* KO MDA-MB-231 clone was treated with pevonedistat and cell proliferation was measured using live-cell imaging. Representative graph of 2 independent repeats was shown. **h** MDA-MB-231 cells were treated with 1000, 100 nM pevonedistat or 0.1% DMSO for 24 h and the total ubiquitin was tested using Western Blotting. Representative image of 2 independent repeats was shown. Control MDA-MB-231 cells were treated with 1000, 400, 100 or 10 nM of TAK-243 (**i**) or 1000, 100, 10 nM of pevonedistat (**j**) or 0.1% DMSO for 24 h, and NAE1 protein expression was measured using Western Blotting. Representative image of 2 independent repeats was shown. Source data are provided as a source data file.

The role of protein neddylation on cancer immunogenicity has been investigated using NAE inhibitors. Pevonedistat treatment causes proteome instability and strongly potentiates response to ICB antibodies in mismatch repair-deficient (dMMR) colon cancer cells[52]. In glioblastoma models, pevonedistat up-regulates PD-L1 expression on cancer cells and synergizes with ICB antibodies in mice[53]. In our experimental models, genetic deletion of *NEDD8* in human TNBC cells does not alter the expression of HLA-ABC nor PD-L1 but enhances the expression of HLA-DR. In TICS, *NEDD8* KO cells strongly enhance immune activation and result in anti-tumor effects after PD-1 blockade in tumor-bearing mice. Interestingly, blocking the conjugation of NEDD8 protein to substrates by deleting the C-terminus diglycine residues achieves similar immune activation in TICS, but the therapeutic potential of this mechanism remains to be validated in mouse models. Despite the low patient number, *NEDD8* mRNA expression show association to pathologic complete response rates in breast cancer patients receiving chemo-immunotherapy[16]. Because NEDD8 is widely expressed by many cell types in the tumor micro-environment, single-cell RNA sequencing datasets in a large cohort of TNBC patients are needed to validate the clinical relevance of our findings.

When exposed to primary human T or NK cells in vitro, NAE inhibitors strongly dampen cell proliferation activated through the CD3/28 pathway at comparable potencies to human TNBC cancer cells. Pevonedistat at intermediate concentrations enhance immune activation primed by nivolumab, but the effect diminishes at high compound concentrations in TICS. These observations are in line with published results, where neddylation inhibitors block TCR signaling[54,55] and anti-bacterial T cell immunity[56]. Because these compounds do not directly target NEDD8 and exert inhibitory functions on protein ubiquitination, the precise mechanistic insights of NEDD8 in immune cell activation and homeostasis should be further dissected using genome editing tools.

Notably, a phase 1 clinical trial combining pevonedistat and pembrolizumab has been performed in mismatch repair deficient colon cancer patients (NCT04800627). It is reasonable to hypothesize that metronomic or intermittent dose-scheduling, as well as targeted delivery[57] of these compounds to tumor lesions could improve the therapeutic index in combination with immunotherapy[20]. In light of the unique functions of NEDD8, modalities that directly limit NEDD8 expression, e.g., RNAi or CRISPR-based therapeutics, could mitigate negative effects of current pharmacological inhibitors on the immune system.

In summary, we have demonstrated that the *NEDD8* gene is a vulnerability to ICB drugs in TNBC. Deficiency of NEDD8 protein in TNBC cancer cells alters immunogenicity that leads to potent immune response after ICB therapy. However, the detailed molecular mechanisms linking NEDD8 loss and enhanced immunotherapy response remain to be investigated. Given that NEDD8 is a key regulator for the post-translational network, modification of immune

checkpoint receptors or ligands should be characterized. Further, our data uncover mechanistic insights of protein neddylation and gene essentiality. A direct and optimized targeting approach against the NEDD8 protein could pave the way to the development of next generation immunotherapy strategies in TNBC and beyond.

## Methods
Details of all antibodies, reagents and oligonucleotide sequences can be found in Supplementary Tables 1–3. The number of detected proteins in proteomics analysis was shown as Supplementary Table 4.

### Study approval
All animals were housed at the animal facility at the Department of Immunology, Genetics and Pathology in the Rudbeck laboratory at Uppsala University, and all studies were approved by the Swedish Board of Agriculture at Jönköping, Sweden (Dnr: 5.8.18-06394/2020).

Buffy coats from healthy donors were obtained from the Uppsala University Hospital. Because donors were fully anonymous, no ethical permission was required.

### Animal studies
In order to study the biological effects and therapeutic potential of NEDD8 on tumor growth, NEDD8 KO or control murine breast cancer cells were injected into syngeneic mouse models. Six to ten weeks old female C57BL/6NTac or C57BL/6J mice were purchased from Taconic. All mice were housed in a barrier facility at the Rudbeck Laboratory (Uppsala University) with a humidity between 45 and 65% and an average temperature of 23 degrees. The dark/light cycle was fixed to 12 h. For EO771 studies, $4–6 \times 10^5$ cells were injected subcutaneously (s.c.) in 100 ul serum free Iscove's Modified Dulbecco's medium (IMDM, Thermo Fisher Scientific). Mice were palpated regularly for tumor detection. Tumor volumes were calculated using the formula $V =$ (length*width^2)/2 and mouse body weights were monitored over the course of the study. The maximal tumor volumes were 1500 mm³. When tumors were palpable or established EO771 tumor-bearing mice were injected intraperitoneally (i.p.) with an anti-PD-1 antibody (clone RMP1-14, BioXcell), or a Rat isotype IgG2a control (clone 2A3, BioXCell) every 3 days (50 µg per mouse). To deplete CD8+ T cells, an anti-CD8a depleting antibody (200 µg, clone 2.43, BioXCell) or an IgG2b isotype control (clone LTF-2, BioXCell) were infused i.p. 4 days after tumor inoculation every 3 days, followed by treatment with 100 µg anti-PD-1 or Rat IgG2a isotype antibody on days 7, 10, 13.

### Human cell lines
Human breast cancer cell line, MDA-MB-231 (92020424, Sigma Aldrich), and HEK293T cells (CRL-3216, American Type Culture Collection, ATCC) were purchased. HCC1937 and BT549 cell lines were a gift from Dr. Óscar Fernández-Capetillo (Karolinska Institutet, Sweden). Mouse breast cancer cell line EO771 was kindly provided by Dr.

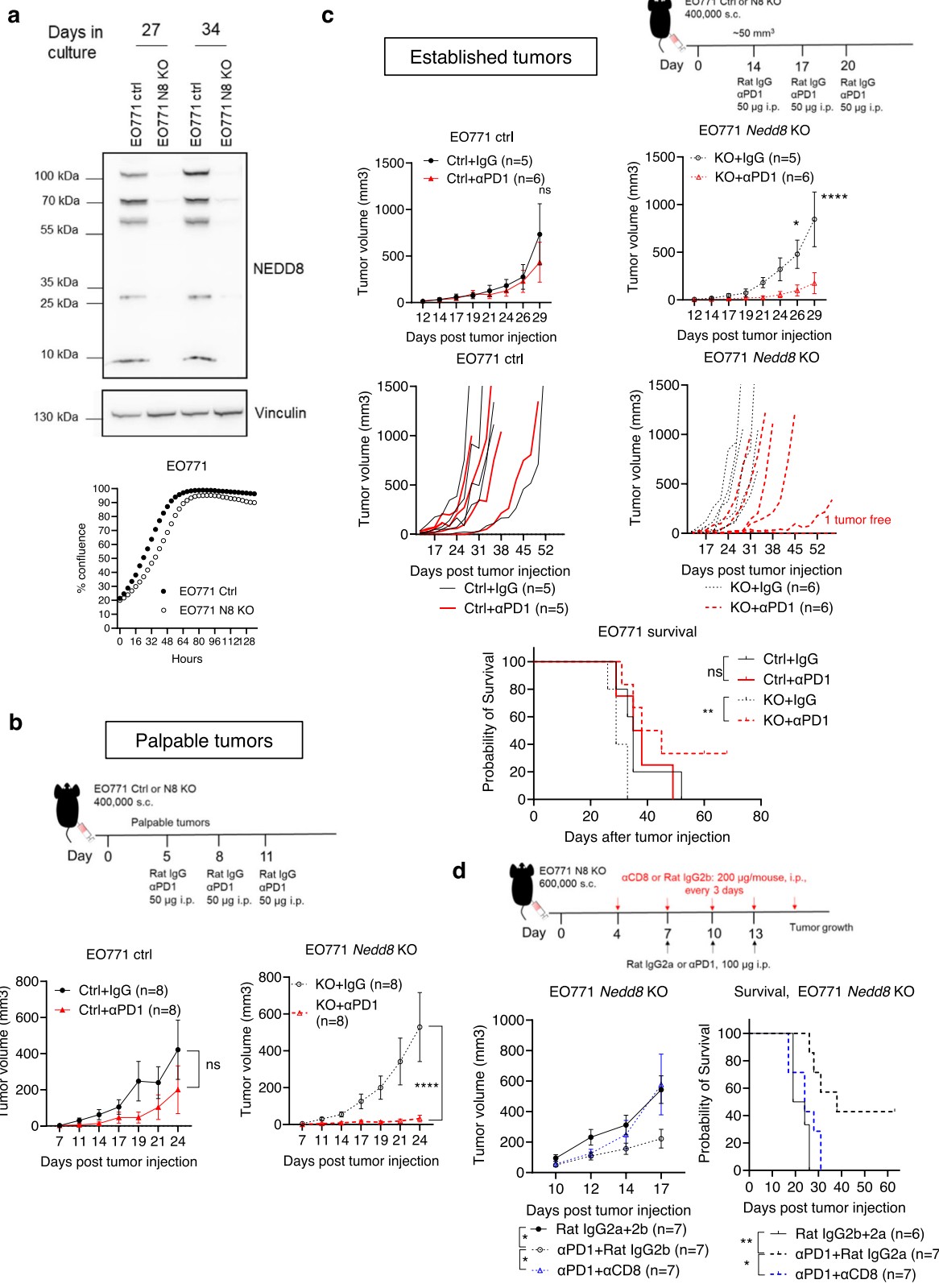

Maria Ulvmar (Uppsala University, Sweden). Unless otherwise stated, all cell lines were maintained in IMDM medium (Thermo Fisher Scientific) containing 10% heat-inactivated Fetal Bovin Serum (FBS) and 1% penicillin-streptomycin solution (Thermo Fisher Scientific) at 37 °C with 5% carbon dioxide. Cell lines were authenticated using DNA fingerprinting (Eurofins) and checked for mycoplasma infection routinely (MycoAlert, Lonza).

## Isolation of human primary immune cells

Buffy coats from healthy blood donors were received from the blood center at the Uppsala University Hospital, Sweden. Peripheral blood mononuclear cells (PBMC) were isolated using SepMate tubes-50 (Stem Cell Technologies) by density gradient centrifugation. Briefly, 10 ml Lymphoprep reagent (Stem Cell Technologies) was added to the tubes followed by addition of blood on top of Lymphoprep. The tubes

**Fig. 6 | Anti-tumor effects of PD-1 blockade on *Nedd8* deficient breast tumors.**
The *Nedd8* gene was deleted using CRISPR/Cas9 in EO771 cells. **a** NEDD8 protein
expression was tested at different passages by Western Blotting and cell pro-
liferation was monitored using a live-cell imaging system. Representative experi-
ment of 3 independent repeats was shown. **b** Four hundred thousand control (Ctrl)
or *Nedd8* knock-out (KO) EO771 cells were injected subcutaneously (s.c.) in 100 µl
medium in 6–10 weeks old female C57BL/6NTac mice. When tumors were palpable,
50 µg of an αPD-1 antibody (RMP1-14) or the Rat IgG2a isotype control (2A3) were
injected intraperitoneally (i.p.) in 100 µl PBS on day 5, 8 and 11 (8 mice per group).
Tumor volumes were compared on day 24. Representative experiment of 3 repeats
was shown. **c** Ctrl or *Nedd8* KO EO771 cells were injected s.c. as above and treatment
began when average tumor volume reached 50 mm$^3$ on day 14, 17 and 20 (at least 5
mice per group). Tumor growth was followed in all mice until the study endpoint.
Survival of the mice was demonstrated in a Kaplan–Meier curve. Representative
experiment of 2 independent repeats was shown. **d** Six hundred thousand ctrl or
*Nedd8* KO EO771 cells were injected s.c. as above. A depletion antibody against
CD8+ T cells (2.43) or the Rat IgG2b isotype control (LTF-2) was injected i.p. in
100 µl PBS every 3 days from day 4 (200 µg per mouse, 7 mice per group). Tumor
growth was compared on day 17 and survival of the mice was demonstrated in a
Kaplan–Meier curve. Representative experiment of 2 independent repeats was
shown. Data were shown as mean ± SEM. Statistical differences on the tumor
volumes were determined using unpaired two-tailed *T*-test and survival differences
were calculated using Kaplan–Meier curves and a log-rank test (Mantel–Cox).
*$p < 0.05$; **$p < 0.01$; ****$p < 0.0001$. Source data are provided as a source data file.

were then centrifuged at 1200×g for 10 min. Next, cell suspension
above the Lymphoprep was collected and PBMCs were washed twice
with phosphate-buffered saline (PBS, Thermo Fisher Scientific). For
optimal lysis of red blood cells, 5 ml ACK lysis buffer (Thermo Fisher
Scientific) was added to the cells and incubated in the dark for 10 min
at room temperature followed by centrifugation at $500 \times g$ for 5 min.
After that, primary monocytes were removed by an EasySep CD14+
selection kit II (Stem Cell Technology) according to the manufacturer's
instructions. Primary human lymphocytes were stored in −150 °C
until use.

### Deletion of individual genes using CRISPR/Cas9
To delete genes of interest, ribonucleoprotein (RNP) complexes con-
taining gRNAs, i.e., crRNA + tracrRNA, targeting human or mouse
genes (Supplementary Table 3) were introduced into cancer cells using
the Neon transfection system (Thermo Scientific). Briefly, 1 µl crRNAs
(100 µM), 1 µl trancrRNA (100 µM) and 1.7 µl nuclease free duplex
buffer (IDT) were added to a PCR tube to form the RNP complexes. A
negative control reaction was set up without the crRNA sequence
(referred as ctrl cells). PCR tubes were then boiled at 95 °C for 5 min
and cooled down at 4 °C. Then, Cas9 endonuclease (10 mg/ml, IDT)
was added to the reaction, followed by incubation at room tempera-
ture for 15 min. A carrier DNA sequence (100 µM) was then added to
the tubes at a final volume of 0.3 µl. Subsequently, the Neon trans-
fection system was prepared according to the manufacturer's
instructions, cell pellets ($5 \times 10^5$ cells) were resuspended in 5 µl resus-
pension buffer R or buffer T and mixed with the same volume of the
RNP complex. Immediately after, the cell mixture was loaded into the
neon pipette tips and the electroporation process was then run using
specific programs on a Neon transfection system. Transfected cells
were cultured and incubated at 37 °C with 5% carbon dioxide until use.
To achieve complete gene deletion, gene-targeting or control RNP
complexes were repeatedly transfected to cells.

### Generation of MDA-MB-231 cells expressing the truncated NEDD8 protein (NEDD8-T)
The plasmid pHAGE-EF1-dCas9-KRAB (Addgene, a kind gift of Scot
Wolfe) was digested with BsrGI (New England BioLabs) and the back-
bone was gel purified. Gibson assembly was used to insert a gBlock
(IDT) containing Gibson arms, a Kozak sequence and coding for a
truncated version of the NEDD8 protein (NEDD8-T) that lacked the
C-terminus diglycine residues (Supplementary Table 3). The resulting
plasmid was sequence verified by Sanger sequencing. For lentivirus
production, $5 \times 10^6$ HEK293T cells were seeded in a T175 tissue culture
flask and transfected with the cargo plasmid as well as packaging
plasmids psPAX2 (Addgene) and pCMV-VSVG (Addgene) using serum
free medium Opti-MEM and transfection reagent Fugene 6 (Promega).
Virus containing medium was collected after 48 h, filtered and 40-fold
concentrated using the lenti X concentrator (Takara bio). Virus was
pelleted by centrifuging at $1500 \times g$ for 45 min at 4 °C and resuspended

in sterile DMEM + 1% BSA. The functional titer of the library virus was
estimated from the fraction of puromycin resistant cells after trans-
duction with different amounts of virus using serial dilution method.
A low MOI of 0.2 was selected for the transduction of NEDD8 KO MDA-
MB-231 cells followed by puromycin selection at 2 µg/ml. The expres-
sion of NEDD8 was analyzed by western blotting.

### Western blotting
Cell lysates were prepared for western blot analysis using antibodies
against NEDD8, UBE2T, NAE1 and GPX4 (Supplementary Table 1). In
brief, cell pellets were lysed in RIPA buffer without additional reducing
reagents (1 mM EGTA, 20 mM Tris, 150 mM NaCl, 1 mM EDTA, 1% NP-
40, 1 mM NaF, 1 mM NaVO3, 1 mM sodium phosphate) with protease
inhibitor cocktail (Thermo Scientific) on ice for 15 min, followed by
centrifugation at $17,000 \times g$ for 12 min/4 °C to remove debris.
According to the manufacturer's instructions, protein concentrations
were determined by the Bicinchoninic Acid (BCA) Assay (Thermo
Scientific). After that, the SDS loading dye-treated proteins were boiled
at 70 °C for 10 min and separated by 4–12% SDS-PAGE gel (Invitrogen),
transferred to nitrocellulose membrane (Invitrogen). The membranes
were blocked with 5% nonfat SKIM milk powder (OXOID), followed by
the addition of primary antibodies and incubation at 4 °C overnight.
On the following day, either anti-mouse or anti-rabbit IgG HRP-linked
secondary antibody (Cell Signaling Technology) was added to the
membranes at room temperature for 1 h. Bands were visualized using
super signal west pico plus or west femto chemiluminescent substrate
(Thermo Scientific) and Amersham Imager 680 machine (GE Health-
care). After each step the membranes were washed with TBST (1X TBS,
0.05% Tween 20, dH$_2$O). Vinculin, β-Actin or GAPDH were used as a
loading control.

### Live imaging for cell proliferation
In order to analyze the effects of different inhibitors on breast cancer
cell proliferation in real time, the incucyte zoom live imaging system
was used. Triple-negative breast cancer cells were seeded at $5 \times 10^3$
cells in a 96-well flat bottom plate and incubated overnight to allow
tumor adherence. Tumors cells were then treated with inhibitors at
indicated doses or 0.1% DMSO (control) in 100 µl of growth medium.
The plate was then incubated into the incucyte image system at indi-
cated time points to evaluate cancer cell proliferation. The cell con-
fluence proportion of inhibitor-treated or DMSO-treated cells was
plotted against the time. Inhibitor concentrations were log$_2$
−transformed and the half-maximal inhibitory concentration (IC50)
value was calculated for each cell line using GraphPad software.

### Generation of pevonedistat-resistant cancer cells
Pevonedistat-resistant MDA-MB-231 cells were derived from original
parental cell line by continuous exposure of pevonedistat in vitro
(MedChemExpress). Briefly, MDA-MB-231 WT cells ($5 \times 10^5$) were see-
ded in a 6 well plate and allowed to adhere overnight at 37 °C. Then, the

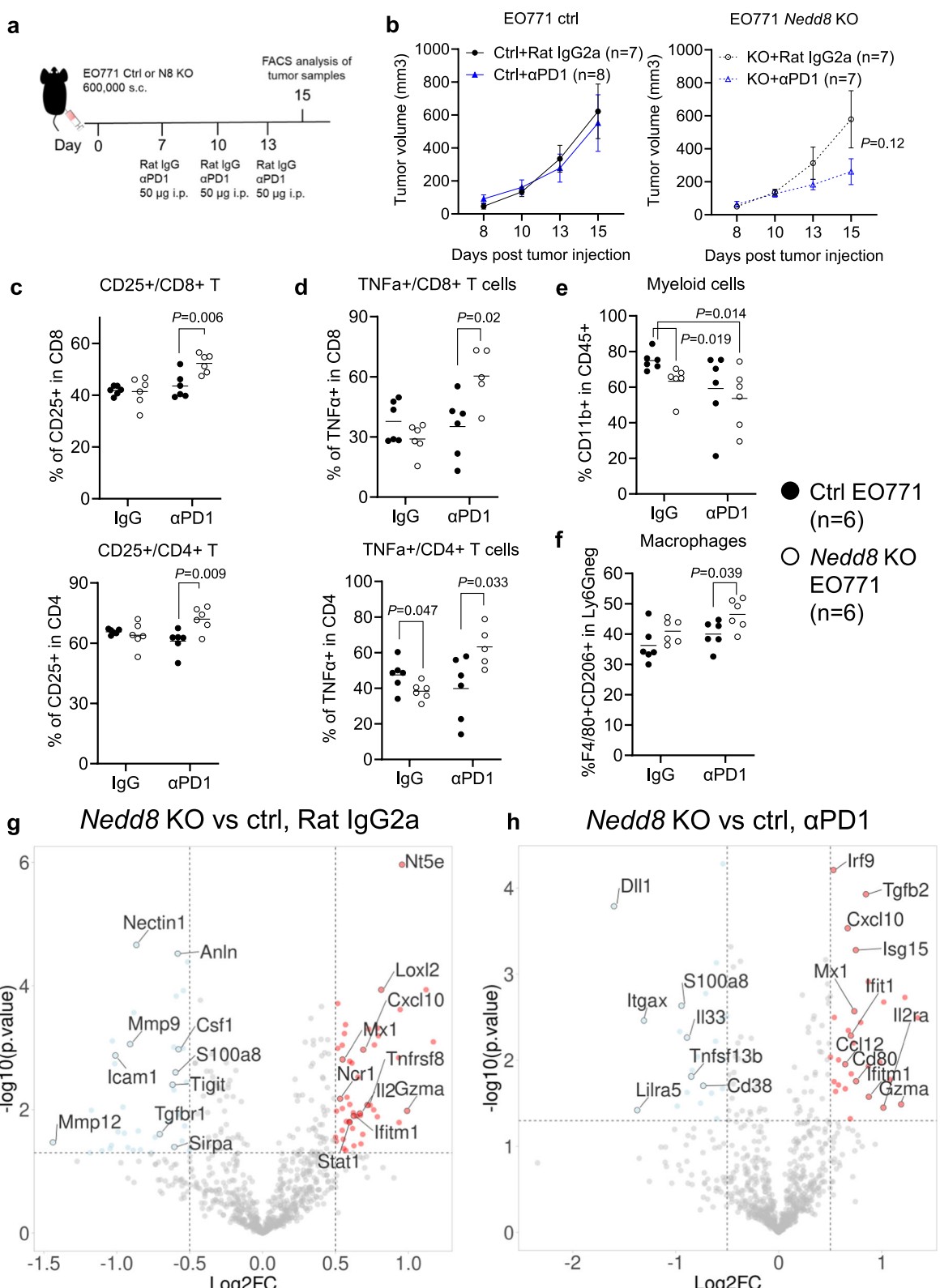

**Lymphocytes proliferation assay**

cells were treated with 500 or 1000 nM of pevonedistat and sub-cultured upon reaching 65–70% confluency. At this time point the media was removed and the above process was repeated. This development period carried out for ~3 months. The sensitivity of resistant cells to pevonedistat was determined using the incucyte zoom live imaging system, as described above. This resistant subline was stored in −80 °C until use.

Primary human lymphocytes were isolated from healthy donors and labeled with the CTV dye as mentioned above. Lymphocytes (1 million cells/ml) were seeded in a 96-well flat bottom plate and activated with cult anti-CD3/CD28 beads (0.4 μl/well, Stemcell) ± rhIL2 (100 ng/ml, Peprotech). Inhibitors of the neddylation pathway, i.e., pevonedistat or TAS4464 were added in 0.1% DMSO at different concentrations and

**Fig. 7 | Immunological changes induced by PD-1 blockade in *Nedd8*-deficient tumors. a** Six hundred thousand control (ctrl) or *Nedd8* knock-out (KO) EO771 cells were injected subcutaneously (s.c.) in 100 µl medium in 6–10 weeks old female C57BL/6NTac mice. On day 7, 10 or 13, 50 µg of an αPD-1 antibody (RMP1-14) or the Rat IgG2a isotype control (2A3) were injected intraperitoneally (i.p.) in 100 µl PBS (7 or 8 mice per group). **b** Tumor volumes were recorded until day 15, when cells were harvested for analysis using flow cytometry (6 tumors per group). Data were shown as mean ± SEM and tested using unpaired two-tailed *T*-tests. Percentages of (**c**) CD25+ T cells, (**d**) TNFα+ T cells, (**e**) CD11b+ myeloid cells or (**f**) macrophages were

compared among groups. Each dot represented an individual tumor and the average values were shown, unpaired two-tailed *T*-test. Tumors were harvested from an independent in vivo study with the same design and mRNA samples were isolated from tumors and quantified using a Nanostring immuno-oncology panel. Differentially expressed genes were shown when comparing (**g**) *Nedd8* KO and control tumors treated with the isotype control antibody, or (**h**) treated with the PD-1 blockade using a cut-off of Log2 fold changes > 0.5 and *p* values < 0.05, unpaired two-tailed *T*-test. Source data are provided as a source data file.

incubated for 4 days. Effects of inhibitors on lymphocyte proliferation and surface markers (Supplementary Table 1) were analyzed by flow cytometry on the CytoFlex instrument.

## Tumor-immune co-culture system (TICS)

To set up the TICS assay, triple-negative breast cancer cells were harvested following the standard protocol for passaging adherent cells. Next, up to 10,000 cancer cells per well were seeded in a 96-well flat bottom plate in 100 µl cell culture medium. The plate was incubated overnight to allow cell adherence. On the next day, healthy donor-derived primary human lymphocytes were incubated in PBS containing 1.42 nM CellTrace Violet dye (CTV, Thermo Fisher Scientific) and incubated in the dark for 10 min. After washing twice with PBS, lymphocytes (3 million cells/ml) were added to the tumor-loaded plate in 100 µl culture medium. FDA-approved checkpoint inhibitors, nivolumab (Bristol-Myers Squibb) or durvalumab (AstraZeneca) were added to the TICS plate at a final concentration of 10 µg/ml in order to inhibit the PD-1/L1 pathway.

For the inhibitor treatment studies, pevonedistat was added to the tumor-immune co-culture plate at indicated doses or 0.1% DMSO (control) on day 3 after co-culture. After 5 days incubation, release of IFN-γ and granzyme B were quantified by ELISA in culture supernatants. In some experiments proliferation and surface protein expression of different immune cell subsets were analyzed by flow cytometry using a CytoFlex S or LX instrument.

## Flow cytometry analysis

For in vitro assays, CTV-treated lymphocytes were harvested from TICS assay and transferred to a 96-well V bottom plate. The cells were centrifuged at $700 \times g$ for 4 min, followed by washing them twice with PBS. After that, cell pellets were resuspended in 20 µl PBS containing aqua fixable live/dead marker (Thermo Fisher Scientific) and then incubated at room temperature for 15 min. The cells were then washed twice with PBS and resuspended in 20 µl master mix containing detection antibodies for surface markers. After 20 min incubation at 4 °C, the cells were washed and resuspended in 150 µl PBS for analysis. To determine the expression of immune related surface markers on NEDD8 KO and control cells, a multi-color flow cytometer was used. In brief, triple-negative control or NEDD8 KO breast cancer cells ($5 \times 10^5$) were cultured in 6 well flat bottom plate in culture medium and incubated overnight to allow cells to attach. Following treatment with ± rhIFNγ (50 ng/ml) for 24 h, cells were harvested, and centrifuged at $350 \times g$ for 4 min. Then, the cells were resuspended in 900 µl PBS and distributed in a 96-well V bottom plate in triplicates (200 µl/well). Subsequently, the plate was centrifuged at $700 \times g$ for 4 min and resuspended in 20 µl PBS containing blue-fluorescent reactive dye (Thermo Fisher Scientific), detection antibodies for surface proteins (1:100) or the matching isotype control IgG for 25 min at 4 °C. After being washed with PBS, the cells were resuspended in 150 µl PBS and transferred into FACS tubes for analysis.

For in vivo studies, single cells from tumor tissues were generated using a Tumor Dissociation Kit (Miltenyi Biotech) using the GentleMacs instrument according to the manufacturer's instructions. Subsequently, cells were loaded in a 96-well V bottom plate, and stained with 20 µl PBS containing an Aqua fixable live/dead marker (1:200) and

a Fc receptor blocking antibody (1:100, Thermo Fisher Scientific). Cells were then washed with PBS and stained with 20 µl PBS containing antibodies for surface proteins (1:100) for 30 min at 4 degrees. To detect intracellular proteins including FoxP3 and TNFa, cells were fixed and permeabilized using a FoxP3/transcription factor staining buffer set (eBioscience) and incubated with fluorochrome-conjugated antibodies (1:50) for 45 min at 4 degrees. The rest of the cells were frozen and stored in −150 °C until use.

All samples were read on Cytoflex S or LX (Beckman coulter), as well as a LSR Fortessa (BD Biosciences) instruments and the data were then analyzed with FlowJo software V10.

**Nanostring analysis.** In order to quantify the mRNA expression of a panel of genes in mouse WT or *Nedd8* KO tumors after anti-PD-1 treatment, mRNA molecules were isolated from single cells using the RNeasy Mini Kit (Qiagen) according to the manufacturer's instructions. Then, the purity of mRNA molecules was determine by the ratio of absorbance at 260 nm and 280 nm. Subsequently, mRNA samples were prepared for nanostring analysis using nCounter immuno-oncology panel.

## Cytokines quantification

Human IFN-γ ELISA kit (Biolegend or MabTech) and granzyme B ELISA kit (MabTech) were used to measure cytokines secretion. Supernatants were collected from TICS, followed by centrifugation at $700 \times g$ for 4 min to remove cell debris. After preparation of samples, ELISAs were conducted according to the manufacturers' protocols. After measuring the absorbance at a wavelength of 450 nm and 570 nm, subtraction of 570 nm readings from those at 450 nm was performed on a CLARIOstar Plus instrument (BMG Labtech), followed by subtraction of an averaged background signal. IFN-γ and granzyme B concentrations were then calculated and plotted against different number of cancer cells using GraphPad software.

## Generation of stable Cas9 expressing cells

Control or NEDD8 CRISPR KO MDA-MB-231 human TNBC cells were lentivirally transduced with pLenti-Cas9-T2A-Blast-BFP to express a codon optimized, WT SpCas9 flanked by two nuclear localization signals linked to a blasticidin-S-deaminase−mTagBFP fusion protein via a self-cleaving peptide (derived from lenti-dCAS9-VP64_Blast, a gift from Feng Zhang, Addgene #61425). Following blasticidin selection, a stable BFP-expressing population was isolated by repeated FACS sorting (Sony SH800).

## Genome-wide CRISPR screens

The genome-wide Brunello sgRNA library[58] was synthesized as 79 bp long oligos (indicated in bold in the sequence below, CustomArray, Genscript). The oligo pool was doublestranded by PCR to include an A-U flip in the tracrRNA[59], 10 nucleotide long random Unique Molecular Identifiers, and an i7 sequencing primer binding site[14].

*ggctttatatatcttgtggaaaggacgaaacaccgnnnnnnnnnnnnnnnnnnnnngtttaagagctagaaatagcaagtttaaataaggctagtccgttatcaacttgaaaaagtggcaccgagtcggtgcttttttGATCGGAAGAGCACACGTCTGAACTCCAGTCACNNNNNNNNNNNNaagcttggcgtaactagatcttgagacaaa*

The resulting PCR product with the sequence was cloned by Gibson assembly into pLenti-Puro-AU-flip-3xBsmBI[14]. The plasmid library was input sequenced to confirm representation and packaged into lentivirus. The functional titer of the library virus was estimated from the fraction of puromycin resistant cells after transduction with different amounts of virus. For the screen, Cas9-expressing target cells were transduced with the library virus in duplicate at an approximate MOI of 0.3 and a coverage of 1000 cells per guide in the presence of 2 µg/ml polybrene. Transduced cells were selected with 2 µg/ml puromycin from day 2 to day 10 post transduction. For the gene-essentiality screen in WT and NEDD8 KO MDA-MB-231 cells were transduced with Brunello library virus and were propagated for 21 days. Cell numbers per replicate were kept at >80 million/replicate throughout to ensure full library coverage.

At the end of cell culture, floating cells were gently washed away and genomic DNA was isolated from cancer cells using the QIAmp DNA Blood Maxi kit (Qiagen). Guide cassettes were amplified by PCR as described[14], using modified primers PCR2_fw *acactcttccctacacgacgctcttccgatctcttgtggaaaggacgaaacac* and PCR3_fw *aatgatacggcgaccaccgagatctacac* [i5] *acactcttccctacacgacgctct*, respectively. The amplicons were sequenced on Illumina NovaSeq, reading 20 cycles Read 1 with custom primer *CGATCTCTTGTGGAAAGGACGAAAC ACCG*; 10 cycles index read i7 to read the UMI, and six cycles index read i5 for the sample barcode.

### Data analysis of genome-wide CRISPR screens

NGS data was analyzed with the MAGeCK software[15] and by UMI lineage dropout analysis[14]. To reduce gene search space, gRNAs targeting mitochondrial and ribosomal genes ($n = 638$) retrieved using the R package biomaRt (MT, rRNA, rRNA_pseudogene and ribozyme biotypes) were excluded. Gene essentiality scores were calculated for each gRNA using MAGeCK for each comparison.

In order to perform enrichment analysis, depleted gRNAs were selected according to the distribution of essentiality scores using a predefined cut-off, i.e., mean-2SD for TICS screens (−0.16 in screen 1 and −0.48 for screen 2) and mean-3SD for the NEDD8 synthetic lethality screen (−1.9 for WT cells and −1.77 for *NEDD8* KO). Next, over-represented pathways were revealed using EnrichAnalyzer function from the MAGeCKFlute R package using the hypergeometric test method.

To compare our NEDD8 synthetic lethality screen to publicly available large scale CRISPR KO screen, we downloaded data of the 2022 Q4 release from the DepMap project [https://depmap.org/portal/download/all/], i.e., gene effects (CRISPRGeneEffect.csv) and cell line metadata (model.csv).

### Proteomics and data analysis

Cell pellets from MDA-MB-231 control and NEDD8 KO cells (4 pellets of each line) were lysed in 100 µl of 1% β-octyl glucopyranoside and 6 M urea containing lysis buffer using a sonication probe for 30 s (3 mm probe, pulse 1 s, amplitude 40%) according to the standard operating procedure. After homogenization, the samples were incubated for 60 min at 4 °C during mild agitation. The lysates were clarified by centrifugation for 10 min (14,000 × *g*). Precipitate from all samples was pressed to get more liquid. The supernatant containing extracted proteins was collected and further processed. The total protein concentration in the samples was measured using the DC Protein Assay with bovine serum albumin (BSA) as a standard. Next, aliquots corresponding to 35 µg of proteins were taken out for digestion. The proteins were reduced, alkylated, on-filter digested by trypsin using 3 kDa centrifugal spin filter (Millipore). The collected peptide filtrate was vacuum centrifuged to dryness using a SpeedVac system. The samples were dissolved in 100 µl 0.1% formic acid and further diluted 4 times prior to LC-MS/MS analysis. The peptides were separated in reversed-phase on a C18-column with 150 min gradient and electrosprayed on-line to a Q-Exactive Plus mass spectrometer (Thermo Finnigan). Tandem mass spectrometry was performed applying HCD.

The RAW-data file was quantitatively analyzed by the quantification software MaxQuant 1.5.1.2. Proteins were identified by searching for proteins from *Homo Sapiens* proteome extracted from Uniprot in February 2020. The search parameters were set to Taxonomy: Homo Sapience, Enzyme: Trypsin. Fixed modification: Carbamidomethyl (C) and variable modifications were Oxidation (M), Deamidated (NQ). 3278 proteins (protein groups) were identified in total in all 8 samples.

Differential protein expression was calculated with R version 4.0.5. Only proteins that were expressed in at least 2 out of 4 replicates of each cell line were considered for statistical analysis. Proteins expressed at least in 3 out of 4 technical replicates of one group and not detected in all 4 technical replicates of the other cell line were considered as uniquely expressed.

A Welch's unequal variances *t*-test was applied to determine differences in expression between proteins expressed in both control and KO. The False Discovery Rate was calculated to adjust the *p* value. An absolute Log2FC above 4 and an FDR below 0.2 were set as thresholds for differentially expressed proteins (DEPs). Subsequently, DEPs and unique proteins were divided into either upregulated in NEDD8 KO or upregulated in control cells for pathway analysis. Proteins were queried for over representation analysis against the Reactome and Gene Ontology Biological Process collections from the Molecular Signature Database using clusterProfiler. Protein interactions were visualized using STRING and Cytoscape V.3.9.1.

### Analysis of published patient dataset

Publicly available sequencing data from breast cancer patients treated with paclitaxel ($n = 179$) or paclitaxel in combination with pembrolizumab ($n = 69$) were retrieved (GSE194040)[16], which was part of the I-SPY2 neoadjuvant platform trial (NCT01042379). Patients in these two arms ($n = 248$) were stratified according to *NEDD8* mRNA expression, using quartiles as cut-off points. Patient subgroups were then annotated by their response to the treatment for further comparisons.

### Statistical analysis

FlowJo V10 software was used to analyze data from flow cytometry analysis. All results were summarized and analyzed using a GraphPad Prism 9 or 10 software. Appropriate statistical analyses were performed using unpaired two-tailed *T*-tests with significance determined at 0.05. Two-Way ANOVA test was used for comparing parameters between multiple experimental groups, as indicated in the figure legends.

### Reporting summary

Further information on research design is available in the Nature Portfolio Reporting Summary linked to this article.

## Data availability

The mass spectrometry proteomics data generated in this study have been deposited in the ProteomeXchange Consortium via the PRIDE[60] partner repository under the identifier PXD051061. The processed proteomics results are included as Supplementary Data 2. Raw data from the Nanostring analysis is included as Supplementary Data 3. The processed gRNA and gene level data are included as Supplementary Data 1. The publicly available large scale cell line CRISPR KO screen data (2022Q4 release) used in this study are available in the Cancer Dependency Map portal (DepMap) [https://depmap.org/portal]. Publicly available breast cancer patient data (NCT01042379) used in this study are available in the NCBI GEO database under accession code GSE194040. The remaining data are available within the Article, Supplementary Information or Source Data file. Source data are provided with this paper.

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

## Acknowledgements

We thank Dr. Maria Ulvmar, Dr. Jordi Carreras Puigvert (Uppsala University, Sweden) and Dr. Óscar Fernández-Capetillo (Karolinska Institutet, Sweden) for sharing key research reagents. We thank Ylva Boström and all the staff at the animal facility for the support of our in vivo experiments. Part of this work was carried out by CRISPR Functional Genomics (CFG), a SciLifeLab funded infrastructure at Karolinska Institutet. The MDA-MB-231 cell line expressing the truncated NEDD8 protein was generated by Soniya Dhanjal at the SciLifeLab CRISPR Functional Genomics unit, Stockholm, Sweden. We acknowledge support from the National Genomics Infrastructure, SNIC (project 2017-7-265), and the Uppsala Multidisciplinary Center for Advanced Computational Science (UPPMAX). The BioVis platform of Uppsala University was used to conduct experiments using flow cytometry, supported by Dirk Pacholsky and staff. The proteomics quantifications were performed by the Mass Spectrometry Based Proteomics Facility (Uppsala University, Sweden), by Dr. Ganna Shevchenko and Prof. Jonas Bergquist. Nanostring analysis was performed at the KIGene core facility at the Karolinska Institute in Sweden. The authors also appreciate the experimental contributions by Ms. Myra Almén and Dr. Yuezhi Chen and assistance on the Incucyte live-cell imaging by Prof. Tobias Sjöblom, Prof. Sven Nelander and Dr. Cecilia Krona. We thank the scientific comments from Dr. Nina Eissler on the manuscript. Y.M.'s research group and the research in this study are generously supported by grants from the SciLifeLab Fellows Program (SLL2019/9), Swedish Cancer Society (220474JIA), Swedish Childhood Cancer Foundation (TJ2019-0057), Swedish Foundation for Strategic Research (FFL21-0043) and Swedish Research Council (2022-01461). M.P.M. and G.K. are supported by the European Union's Horizon 2020 Research and Innovation Programme (Grant agreement No. 950293 - COMBAT-RES).

## Author contributions

Y.M. and I.P. initiated the project and formulated the research hypothesis. I.P., D.N. and Y.M. performed genome-wide CRISPR screens and contributed to data interpretation and analysis. I.P. performed the majority of in vitro experiments and all in vivo experiments. I.P. and D.N. performed the analysis of cells from tumor-bearing mice using flow cytometry and I.P. analyzed flow cytometry results with the support of Y.M. L.P.A. performed in vitro experiments during the revision. M.R.B. performed the analysis of proteomics results. G.K. analyzed and interpreted data from genome-wide CRISPR screens and the public patient dataset. M.P.M. supervised the data analysis and provided guidance for the project development. All authors contributed to the writing and revising of the manuscript.

## Funding

## Competing interests

Y.M. and M.P.M. were former employees of AstraZeneca and hold company shares. Y.M. received funding from Bayer Pharmaceuticals and Novo Nordisk Foundation for unrelated projects. M.P.M. receives funding from Roche and GSK for other projects. Other authors declare no conflict of interest.
