## [Peer Review File · Nature Communications]

Loss of NEDD8 causes vulnerability to immune checkpoint blockade in triple-negative breast cancerREVIEWER COMMENTS

Reviewer #1 (Remarks to the Author):

The ability to inhibit the immune checkpoint blockade (ICB) between tumor cells and the immune system has revolutionized the treatment of cancers that respond to this approach. However, immunotherapy treatment for triple-negative breast cancer (TNBC) has had limited success, despite the fact that most TNBC tumors express PD-L1. Thus, it is essential to identify new ways to sensitize TNBC tumors to ICB blockade.

In their manuscript, Papakyriacou et al. describe the therapeutic approach of inhibiting NEDD8 activation/enzyme (NAE) to treat TNBC by augmenting ICB therapy. The central conclusion of this manuscript is that inhibition of NAE by pevonedistat is synergistic with ICB therapy in a mouse TNBC model, as demonstrated by the animal studies described in Figure 6.

While the authors have pointed out in the discussion that two previous studies (ref 42 and 43) demonstrated the synergy between pevonedistat and ICB therapy with different cancer models, the novelty and significance of this manuscript are that this synergy is also valid for TNBC, which warrants publication in Nat. Comm. However, the experiments described in the first five figures are not compelling, and some of the experiments are "filler experiments" that provide little support to the central conclusion.

Below are the points that the authors must address to make this manuscript better:

1. Figures 1C, D, E, and F make the same point that nine genes are outliers from the CRISPR screen. There is no need to show the same conclusion with four different panels.

Additionally, the number of genes highlighted as outliers in 1D and 1E is not consistent with Fig 1C and 1F and what is described in the text and figure legend.

2. The experiments described in Figure 2 (the generation of NEDD8 KO lines and their characterization) have no relevance to the central conclusion of this study. Moreover, the fact that NEDD8 knockout lines show similar sensitivity to pevonedistat once they have adapted is unexpected, and the authors fail to provide a reasonable explanation or

hypothesis for this unexpected result.

3. The proteomic survey described in Figure 3 to identify protein changes in response to NEDD8 knockout adds very little to the central conclusion of the manuscript.

4. The changes in proteins in response to NEDD8 knockout, as detected by mass spectrometry, show only a 2-4 fold change (Figure 3A). Based on general experience known in analyzing MS/proteomic data, a 2 to 4 fold change is very unlikely to reflect real differences in protein level and must be validated by western blot. For the UBE2T western blot validation (Figure 3D), it is not apparent from Figure 3 or the text how much UBE2T changed by MS.

5. For Fig 3B and 3C (the graphical representation of the pathways affected by NEDD8 knockout), the idea that protein neddylation may regulate these pathways is not new and these two panels provide no new insight into the biology of NEDD8.

6. In Figure 3E, the change in CD55 by MS is approximately a 4-fold decrease in response to NEDD8 knockout. However, by flow cytometry, the difference is only about a 30% decrease in CD55 levels. This major discrepancy highlights the prudence needed in interpreting MS results and the importance of validating protein changes by western blot.

7. In Figure 4A, a 2-fold increase in the intensity of the MS signal is not significant, regardless of the statistics.

Reviewer #2 (Remarks to the Author):

Papakyriacou and colleagues performed a genome-wide screen using CRISPR/Cas9 to identify genes that their deletion sensitizes the response to immune checkpoint blockade therapy (ICB). By designing the so-called Tumour Immune co-Culture System and using a Triple Negative Breast Cancer model cell system the authors identified NEDD8 as a vulnerability gene to ICB treatment. Further in vitro analysis and in vivo testing showed that NEDD8 deletion has strong immunogenic and anti-tumour effects.

Overall, the study is rather interesting. The manuscript is well written (a few spelling mistakes present) and the presented data are of very high quality that support the major claims of the authors.

I feel that there are 2 parts in the manuscript: The first is related to the intriguing finding that NEDD8 deletion causes an adaptation/re-programming process that allows the selection of viable cells. Additionally, the finding that inhibition of the NEDD8 pathway still sensitizes NEDD8 KO cells, raises interesting hypothesis for the role of the NEDD8 conjugating enzymes. The second part is indeed the in vivo validation for the NEDD8 inhibition as an approach to potentiate the ICB response.

While both parts present interesting and exiting findings, mechanistic details are in rather preliminary stage. I think the authors should improve this aspect of the manuscript-at least for one of the 2 parts.

Detailed comments:

Based on the finding that the NEDD8 KO cells are sensitive to the inhibition of NEDD8 enzymes, the authors suggest that the NAE/Ube2M enzymes have additional functions to protein NEDDylation. This in theory is possible but it is a rather strong statement, especially in the absence of supporting data. The link/cross-talk of NEDD8 with other members of the family especially with ubiquitin is well established (Lobato et al., 2021, doi:

10.1016/j.celrep.2020.108635, Maghames et al.2018, doi: 10.1038/s41467-018-06365-0,

Hjerpe et al., 2012, doi: 10.1042/BJ20111671, Leidecker et al., 2012 doi:

10.4161/cc.11.6.19559). Conditions that reduce free ubiquitin allow NEDD8 to engage in the ubiquitin system (atypical neddylation). What happens to ubiquitin in the absence of NEDD8 is not known. Early structural studies indicated that the NAE is rather specific in activating NEDD8 but not ubiquitin. However, I think it is critical to monitor the thioester state of NAE and especially Ube2M in NEDD8 KO cells.

1. Are these enzymes loaded with ubiquitin instead, when NEDD8 is absent?
2. Compared to control cells are the NEDD8 KO cells equally or more sensitive to ubiquitin E1 inhibitors?
2. Could NAE/Ube2m still activate cullin-ring-ligases (CRL) through cullin ubiquitination in NEDD8 KO cells? Monitor the levels of established CRL substrates. The proteomic analysis

shows that the number of proteins with altered stability in NEDD8 KO cells is rather small, relatively to what was expected if CRL dependent degradation was blocked, indicating that CRLs may be still active. Details in the proteomic analysis should be included, for example total number of identified/quantified proteins/peptides.

3. NEDD8 inhibitors require the presence of NEDD8 and the activity of NAE to operate. So how do they work in NEDD8 KO cells? Can they still interact with NAE? The observation that both NAE/Ube2M genes were selected in the screen in NEDD8 KO cells suggest that the inhibitors (Pevonedistat/TAS4464) effect are NAE dependent and not off-target effects. As a comment, TAS4464 is not as specific compared to Pevonedistat-IC50 for the Ubiquitin E1 enzyme is 500nM and 1500nM respectively. This may become important when assessing their in vivo effects at high (uM) used doses. Monitoring their effect on total ubiquitination is a good control.

4. Could a NEDD8 mutant deficient in conjugation (mutation/deletion of the C-terminus diglycine) rescue any phenotypes in the NEDD8 KO cells?

I find the experiments with the Pevonedistat resistant cells quite intriguing. Previous studies identified mutations within NAE that provides resistance to Pevonedistat. Here, there is no effect on protein NEDDylation upon Pevonedistat treatment, hence the resistance. These results also provided the proof-of principle that NAE is indeed the target of these inhibitors through which induce their cytotoxic effects.

In the current study though, Pevonedistat still reduces NEDDylation suggesting that the resistance has nothing to do with the NEDD8 pathway per se. Sequencing may be informative. The increase in the levels of NEDD8 is rather modest-at best-so I feel the claim that the effects in IFN γ and GzmB levels (Fig. 5G,H) are due to increase in NEDD8 are not fully supported. So, overall, I do not feel this is an appropriate system to assess the role of NEDD8 increase in ICB response.

The authors may consider the knockout of the NEDP1 deneddylating enzyme that results in a dramatic increase in protein neddylation and it is not an essential gene, as a more appropriate approach.

It is advised that for the presentation of the statistical analysis on the graphs, each replicate experiment is shown as different colour or symbol to be directly compared with the relevant

replicate in different used conditions.

Reviewer #3 (Remarks to the Author):

Here the authors present 2 genome-wide CRISPR screens in their co-culture system with triple negative breast cancer cells. They found that MDA-MB-231 cells can compensate for NEDD8 loss and these deficient cells are vulnerable to nivolumab, presumably due to enhanced protein expression for antigen presentation (HLA-DRA, -DRB and CD74). Overall the paper is well organized and provides some original findings.

Major conceptual comment:

Targeting genes with CRISPR KO to sensitize immunotherapy is not new. The context of many previous studies should be carefully incorporated.

The study is far from clinical impact because there's no specific inhibitors for NEDD8 and toxicities are unclear.

Technical comments:

The authors need to address the following points:

- 1) The finding that the MDA-MB-231 cancer cells can compensate for loss of an essential gene, NEDD8, is an interesting finding, but also raises the question if normal mammary cells can also compensate or if this is a unique finding of the case of triple negative breast cancer?
- 2) The authors show that NEDD8 KO MDA-MB-231 cells show depleted expression of genes involved in DNA repair and telomere maintenance. Have the authors evaluated DNA damage in these KO cells and their sensitivity to cytotoxic chemotherapy or radiation? This could also increase the clinical impact of their paper.
- 3) For the following results: "Moreover, we observed that gRNAs targeting the NAE1 gene were depleted in both the WT and KO cells in the genome-wide screens (Figure 2C). NAE1 144 encodes the NEDD8-activating enzyme E1 subunit 1 (NAE1), which is a key subunit of the 145 first heterodimer enzyme of the neddylation pathway [16]. Pharmacological inhibitors against 146 NAE, i.e. pevonedistat [25] and TAS4464 [26], have been developed and tested in patients as 147 potential anti-cancer therapies [25, 27]. Cell proliferation assays showed that the WT and KO 148 cells were equally sensitive to pevonedistat (Figure

2E) and TAS4464 (Figure S3D). Deletion 149 of NEDD8 did not influence the expression of NAE1 protein (Figure S3E)."

Isn't this an unexpected finding given that the KO cells can survive without NEDD8 and NAE1 functions in this pathway. Does NAE1 have other functions besides NEDD8 activation which are essential for cells? This would be crucial to explain for the readers.

4) For the following results: "To test whether NEDD8 KO TNBC cells can induce stronger immune cell activation, WT or 194 NEDD8 deficient MDA-MB-231 cells were co-cultured with primary human lymphocytes in 195 TICS +/- ICB drugs. The KO cells induced only marginal enhancements of soluble IFN γ and granzyme B in culture supernatants, as compared to the WT cells (Figure 4C). In contrast, we observed significantly increased production of these immune-activating cytokines in the 198 presence of nivolumab or durvalumab (Figure 4C; Figure S4B). In accordance, absence of NEDD8 in MDA-MB-231 cells induced stronger proliferation of CD8 $^{+}$ and CD4 $^{+}$ T cells in 200 TICS in response to blockade of either PD-1 or PD-L1 (Figure 4D). The activation of NK cells by NEDD8 KO TNBC cancer cells showed only a trend of increase (Figure S4C). The increased release of soluble granzyme B in response to nivolumab or durvalumab was further confirmed using an additional WT/KO cell line pair derived from HCC1937 cells (Figure 4E)."

Do the authors know why they did not see soluble IFN γ and granzyme B in co-culture supernatants of KO cells in the absence of PD1 blockade? Have they profiled their T cells for exhaustion markers that would explain this?

5) The authors have done excellent mouse studies showing the NEDD8 delayed growth of E0771 tumors in B6 mice only upon PD1 blockade. Did they analyze tumor infiltrating lymphocytes in NEDD8 KO tumors and WT tumors with and without PD1 blockade and see increased proliferation and release of soluble IFN γ and granzyme B?

6) The impact and significance of their finding could be increased if the authors analyzed publicly available immunogenomic patient datasets to see if increased expression of NEDD8, or the neddylation pathway in general, is predictive of poorer response to immunotherapy. Could this be used as a biomarker?

Minor points:

In paragraph 2 of the results section please mention how long the cells were co-cultured. I also could not find this information in their methods section. This would allow people to reproduce their results.

Figure 6D- Please show WT cells in the same plot under the same conditions

Sup Figure 1E- It is hard to see a difference in media color between conditions. Were total cell numbers counted?

Sup Figure 4B- Images are too small to see.

POINT-TO-POINT RESPONSE TO REVIEWER COMMENTS

Reviewer #1 (Remarks to the Author):

In their manuscript, Papakyriacou et al. describe the therapeutic approach of inhibiting NEDD8 activation/enzyme (NAE) to treat TNBC by augmenting ICB therapy. The central conclusion of this manuscript is that inhibition of NAE by pevonedistat is synergistic with ICB therapy in a mouse TNBC model, as demonstrated by the animal studies described in Figure 6.

While the authors have pointed out in the discussion that two previous studies (ref 42 and 43) demonstrated the synergy between pevonedistat and ICB therapy with different cancer models, the novelty and significance of this manuscript are that this synergy is also valid for TNBC, which warrants publication in Nat. Comm. However, the experiments described in the first five figures are not compelling, and some of the experiments are "filler experiments" that provide little support to the central conclusion.

Authors' response: We thank the reviewer for the comments. We are sorry if the key message of the manuscript was unclear. Our data revealed an unexpected finding, where existing inhibitors against protein neddylation, e.g. pevonedistat, perturbs broad biological mechanisms beyond protein neddylation. This was exemplified by its clear negative impact on T and NK cells (**Figure 5B**) and inhibition of cells that do not express NEDD8, i.e. deficiency in protein neddylation (**Figure 2E**). To validate this hypothesis, we performed additional experiments during the revision and showed that the compound can inhibit cells lacking *NAE1*, which is the putative target (**Figure 5F and G**), or cells harbor a non-functional NEDD8 protein (**Figure 4F and S5D**). Further investigations revealed that pevonedistat could inhibit protein ubiquitination (**Figure 5H-J**). We therefore performed *in vivo* studies presented in **Figure 6** with the NEDD8 CRISPR KO cells, instead of using pevonedistat. This discovery was rather unexpected and provides new insights into optimizing neddylation inhibitors for cancer therapy. We have now added a new section in the results in line 251-273 to highlight these new data. We also revised the text in the discussion to more clearly demonstrate the importance and novelty of our work in line 384-393 and 412-417.

1. Figures 1C, D, E, and F make the same point that nine genes are outliers from the CRISPR screen. There is no need to show the same conclusion with four different panels. Additionally, the number of genes highlighted as outliers in 1D and 1E is not consistent with Fig 1C and 1F and what is described in the text and figure legend.

Authors' response: We thank the reviewer for this point and comment on the inconsistency of the gene names. The gene names have been unified in the revised Figure 1. We agree that information shown in this figure can be simplified. We have now moved Figure 1F into **Figure S2D** and updated the information in line 119.

2. The experiments described in Figure 2 (the generation of NEDD8 KO lines and their characterization) have no relevance to the central conclusion of this study. Moreover, the fact that NEDD8 knockout lines show similar sensitivity to pevonedistat once they have adapted is unexpected, and the authors fail to provide a reasonable explanation or hypothesis for this unexpected result.

Authors' response: We thank the reviewer for this point. We can appreciate that validating the deletion of NEDD8 in cancer cells may seem irrelevant to the central message. However, even with CRISPR-based genome editing, deletion of the target protein may not be complete and off-target effects may contribute to observed phenotypic changes. Because *NEDD8* is a widely reported common essential gene that is required for cell survival in >1000 CRISPR screens, we strongly felt that it was necessary to conclusively prove the loss of its essentiality in KO cells using CRISPR screens. In addition, the genome-wide CRISPR screens have provided unbiased results to shed light on mechanisms involved in

the survival of NEDD8 KO cells. In our view, this is a crucial part to demonstrate the validity of our work. This has been clarified in line 360-368.

We agree with the reviewer that the effects of pevonedistat on cells should be further investigated. In the revised manuscript, we showed in additional experiments that pevonedistat could inhibit proliferation of *NAE1* KO cells (**Figure 5G**) or cells harbor non-functional protein neddylation (**Figure 4F** and **S5D**), which are the putative targets for the compound. It could be explained by its off-target inhibition on protein ubiquitination (**Figure 5H-J**). Altogether, our results strongly demonstrate that pevonedistat regulates additional cellular process other than neddylation and its clinical utility should be carefully optimized. We have now added a new section in the results in line 251-273 to highlight these new data.

3. The proteomic survey described in Figure 3 to identify protein changes in response to NEDD8 knockout adds very little to the central conclusion of the manuscript.

Authors' response: We thank the reviewer for this point. Protein neddylation is a key mechanism that governs post-translational modifications in cancer cell proteome. Therefore, we sought to map the global protein changes using proteomics. This dataset is valuable in two ways. Firstly, it revealed enhanced expression of antigen presentation molecules upon NEDD8 deletion, e.g. HLA-DR, which was validated in **Figure 4A** and **B** and supported NEDD8's novel role in cancer immunogenicity. Secondly, we identified altered pathways that further shed light on the compensatory mechanisms upon NEDD8 deletion. Together with data generated from genome-wide CRISPR screens in **Figure 2**, we propose that alternative pathways, such as ubiquitination, DNA replication, are activated to enable cellular reprogramming to enable the survival of *NEDD8* deficient cells. This message has been added to the revised manuscript in line 363-368.

4. The changes in proteins in response to NEDD8 knockout, as detected by mass spectrometry, show only a 2-4 fold change (Figure 3A). Based on general experience known in analyzing MS/proteomic data, a 2 to 4 fold change is very unlikely to reflect real differences in protein level and must be validated by western blot. For the UBE2T western blot validation (Figure 3D), it is not apparent from Figure 3 or the text how much UBE2T changed by MS.

Authors' response: We fully agree that data obtained from MS should be validated individually using other methods. We have performed such validations using either western blotting or flow cytometry. For immunological receptors like HLA-DR, it is more appropriate to quantify the surface expression using flow cytometry. In the case of UBE2T, it was classified as 'unique in KO', which means that these peptides were only detectable in the *NEDD8* KO cells but not in control cells (**Figure 3C**). In our validation experiments using western blotting in **Figure 3D**, we confirmed the enhanced expression of UBE2T in NEDD8 KO cells but the protein was weakly expressed in control cells, which could contribute to the lack of signal in MS-based proteomics. This has been mentioned now in line 189-191.

5. For Fig 3B and 3C (the graphical representation of the pathways affected by NEDD8 knockout), the idea that protein neddylation may regulate these pathways is not new and these two panels provide no new insight into the biology of NEDD8.

Authors' response: We thank the reviewer for this point. Although it is true that NEDD8 deletion is expected to alter protein expression in cells, it was unknown how cellular reprogramming upon NEDD8 deletion could support cell survival and immunogenicity. We believe that data from **Figure 3** provided novel evidence of compensatory mechanisms in post-translational modification upon NEDD8 deletion. The resulted cells demonstrated enhanced immunogenicity through upregulation of HLA-DR molecules, which was validated by flow cytometry in **Figure 4**. Together with results from genome-wide CRISPR screens in control or *NEDD8* KO cells, we could map the functional landscape conferring survival and immunogenicity. We have revised our discussion in line 364-369 and 395-401.

6. In Figure 3E, the change in CD55 by MS is approximately a 4-fold decrease in response to NEDD8 knockout. However, by flow cytometry, the difference is only about a 30% decrease in CD55 levels. This major discrepancy highlights the prudence needed in interpreting MS results and the importance of validating protein changes by western blot.

Authors' response: We appreciate the concern raised by the reviewer. This can be explained by the difference in protein detection methods used. While flow cytometry detects only the extracellular domain of CD55, MS-based methods or western blotting capture global expression of the peptides. Considering that the main biological function of CD55 relies on the extracellular domain, we suggest that surface expression of this receptor measured by flow cytometry is a more suitable readout as compared to western blotting. The text has been modified in line 191-192.

7. In Figure 4A, a 2-fold increase in the intensity of the MS signal is not significant, regardless of the statistics.

Authors' response: We thank the reviewer for raising this viewpoint. In **Figure 4A** and **4B**, we demonstrated the statistically significant up-regulation of HLA-DR as a result of *NEDD8* deletion in human TNBC cells. Although the magnitude of increase seemed to be modest, i.e. 2 folds, we believe that this could have a profound biological function on immune activation. This is because the positive control, i.e. recombinant human interferon gamma (IFNG), induced roughly a 2.5-fold change on surface HLA-DR expression. IFNG is a known potent inducer of immune activation and anti-tumor immunity and therefore we demonstrate that the effects of NEDD8 deletion on cancer immunogenicity are also biologically significant. This point has been added in line 199-201.

Reviewer #2 (Remarks to the Author):

Papakyriacou and colleagues performed a genome-wide screen using CRISPR/Cas9 to identify genes that their deletion sensitizes the response to immune checkpoint blockade therapy (ICB). By designing the so-called Tumour Immune co-Culture System and using a Triple Negative Breast Cancer model cell system the authors identified NEDD8 as a vulnerability gene to ICB treatment. Further in vitro analysis and in vivo testing showed that NEDD8 deletion has strong immunogenic and anti-tumour effects.

1. Overall, the study is rather interesting. The manuscript is well written (a few spelling mistakes present) and the presented data are of very high quality that support the major claims of the authors.

Authors' response: We highly appreciate your positive view on our work. We have now read through the manuscript and corrected typos and grammatical errors. We have included a version of the revised manuscript with these changes highlighted in the text.

2. I feel that there are 2 parts in the manuscript: The first is related to the intriguing finding that NEDD8 deletion causes an adaptation/re-programming process that allows the selection of viable cells. Additionally, the finding that inhibition of the NEDD8 pathway still sensitizes NEDD8 KO cells, raises interesting hypothesis for the role of the NEDD8 conjugating enzymes. The second part is indeed the in vivo validation for the NEDD8 inhibition as an approach to potentiate the ICB response. While both parts present interesting and exiting findings, mechanistic details are in rather preliminary stage. I think the authors should improve this aspect of the manuscript-at least for one of the 2 parts.

Authors' response: We are grateful that the reviewer has identified this challenge. Initially we planned to focus on the immunogenicity aspect of NEDD8, which fits well to our research scope and capability. After discussing with our collaborators, however, we reached the consensus that elucidating the survival mechanisms in *NEDD8* deficient cells, as well as revealing the unknown mechanisms of protein

neddylations are equally important and exciting. We hope that presenting our work in the current form to the scientific community could facilitate further research on both aspects of the pathway.

Detailed comments:

3. Based on the finding that the NEDD8 KO cells are sensitive to the inhibition of NEDD8 enzymes, the authors suggest that the NAE/Ube2M enzymes have additional functions to protein NEDDylation. This in theory is possible but it is a rather strong statement, especially in the absence of supporting data. The link/cross-talk of NEDD8 with other members of the family especially with ubiquitin is well established (Lobato et al., 2021, doi: 10.1016/j.celrep.2020.108635, Maghames et al. 2018, doi: 10.1038/s41467-018-06365-0, Hjerpe et al., 2012, doi: 10.1042/BJ20111671, Leidecker et al., 2012 doi: 10.4161/cc.11.6.19559). Conditions that reduce free ubiquitin allow NEDD8 to engage in the ubiquitin system (atypical neddylation). What happens to ubiquitin in the absence of NEDD8 is not known. Early structural studies indicated that the NAE is rather specific in activating NEDD8 but not ubiquitin. However, I think it is critical to monitor the thioester state of NAE and especially Ube2M in NEDD8 KO cells.

Authors' response: We fully agree with your point of view and highly appreciate your excellent questions as an expert in the protein neddylation field. We thank the reviewer for highlighting these important references, which have been incorporated into our revised manuscript. We have performed further experiments to address the concerns. In light of our additional data, we have revised manuscript regarding mechanisms-of-action for NAE inhibitors. Please see our point-by-point replies below.

4. Are these enzymes loaded with ubiquitin instead, when NEDD8 is absent?

Authors' response: We thank the reviewer for this important question. We have now performed additional experiments to detect the ubiquitin levels in parental and *NEDD8* KO cells by western blotting. In **Figure S4B**, we did not observe clear differences when measuring the global ubiquitin levels in this cell line pair. Of note, we did not detect a band around 25-35 kDa, which would correspond to the molecular weight of ubiquitinated UBE2M. This suggests that UBE2M was ubiquitinated at low levels and was not impacted by *NEDD8* deletion in this cell line pair. This new data is mentioned on line 180-182.

To further investigate the biological mechanisms of NAE1, we have generated a new CRISPR KO clone in MDA-MB-231 cells (**Figure 5F**). When blotting for NAE1, we observed a heavier band at around 70 kDa in the three human TNBC cell lines (**Figure S3E**) and it disappeared in *NAE1* KO cells (**Figure 5F**). Because this band was detected at comparable levels in the WT/*NEDD8* KO cell line pairs (**Figure S3E**), we speculated that this was NAE1 conjugated with ubiquitin. Indeed, when treated with an ubiquitin-activating E1 enzyme (UBA1) inhibitor, PYR41, this band was inhibited (**Figure 5I**). Therefore, we concluded that *NEDD8* deletion did not change ubiquitination of NAE1. We have re-structured this part of the results in line 250-272.

5. Compared to control cells are the *NEDD8* KO cells equally or more sensitive to ubiquitin E1 inhibitors?

Authors' response: We thank the reviewer for this point. We have performed additional experiments to test the effects of a selective UBA1 inhibitor, PYR41, on the proliferation of parental or *NEDD8* KO MDA-MB-231 cells. Similar to literature data, the compound was only active on MDA-MB-231 cells at concentrations higher than 10 μ M. In this experimental setting, we observed that the cell line pair was equally sensitive to the inhibitor. This data has been added to **Figure S7D** and mentioned in line 265-268.

6. Could NAE/Ube2m still activate cullin-ring-ligases (CRL) through cullin ubiquitination in *NEDD8* KO cells? Monitor the levels of established CRL substrates. The proteomic analysis shows that the

number of proteins with altered stability in NEDD8 KO cells is rather small, relatively to what was expected if CRL dependent degradation was blocked, indicating that CRLs may be still active. Details in the proteomic analysis should be included, for example total number of identified/quantified proteins/peptides.

Authors' response: We appreciate reviewer's point on cullin ubiquitination. We have performed additional experiments to measure expression of cullin-1 in control or NEDD8 KO MDA-MB-231 cells. As shown in **Figure S4C**, NEDD8 deletion did not alter cullin-1 expression, nor its ubiquitination. The total number of detected peptides have been included in **Supplementary Table S4**. These numbers were comparable between control and NEDD8 KO cells. As the reviewer correctly pointed out, because ubiquitination of cullin-1 was still detectable in NEDD8 KO cells, it could explain the relatively small changes in proteome. We have discussed this point in line 180-182.

7. NEDD8 inhibitors require the presence of NEDD8 and the activity of NAE to operate. So how do they work in NEDD8 KO cells? Can they still interact with NAE? The observation that both NAE/Ube2M genes were selected in the screen in NEDD8 KO cells suggest that the inhibitors (Pevonedistat/TAS4464) effect are NAE dependent and not off-target effects.

Authors' response: We thank the reviewer for highlighting this point. In order to determine whether pevonedistat carries activity other than inhibiting NAE1, we deleted the NAE1 protein using CRISPR/Cas9 in MDA-MB-231 cells (**Figure 5F**). Similar to the NEDD8 KO cells, pevonedistat strongly inhibited the proliferation of *NAE1*-deficient cells (**Figure 5G**). This data shows that pevonedistat targets pathways in addition to NAE1. We believe that pevonedistat could inhibit protein ubiquitination at high concentrations (please see response to the point below). This data is now mentioned in line 251-256.

8. As a comment, TAS4464 is not as specific compared to Pevonedistat-IC50 for the Ubiquitin E1 enzyme is 500nM and 1500nM respectively. This may become important when assessing their in vivo effects at high (uM) used doses. Monitoring their effect on total ubiquitination is a good control.

Authors' response: This is an excellent comment that we have not considered. In the revised manuscript, we assessed whether pevonedistat could have an effect on ubiquitination. Our results showed that pevonedistat at high concentrations, i.e. >1000 nM, inhibited global ubiquitination in MDA-MB-231 cells (**Figure 5H**). This observation coincides with the negative effects on immune activation in TICS at this concentration (**Figure 5C**). To conclusively prove this point, we treated MDA-MB-231 wildtype cells with pevonedistat and observed complete abolishment of NAE1 ubiquitination in **Figure 5J** (see more information in point 4 above). Therefore, we propose that metronomic dosing of pevonedistat could be beneficial for the immune priming effects by avoiding inhibition of protein ubiquitination. This data has been added in the text, line 258-261 and 283-292.

9. Could a NEDD8 mutant deficient in conjugation (mutation/deletion of the C-terminus diglycine) rescue any phenotypes in the NEDD8 KO cells?

Authors' response: We really appreciate this excellent suggestion. We have now generated a mutant cell line, where a truncated version of the NEDD8 protein (NEDD8-T) was expressed in the MDA-MB-231 NEDD8 KO cells using viral transduction. As shown in **Figure 4F**, the truncated NEDD8 protein was detectable but failed to conjugate to substrates due to the loss of C-terminus diglycine residues. Of note, NEDD8-T cells were equally sensitive to pevonedistat, as compared to NEDD8 KO cells (**Figure S5D**). When tested in TICS, NEDD8-T cells showed strong activation of human immune cells that was comparable to NEDD8 KO cells (**Figure 4G**). This data has been mentioned on line 213-218.

10. I find the experiments with the Pevonedistat resistant cells quite intriguing. Previous studies identified mutations within NAE that provides resistance to Pevonedistat. Here, there is no effect on protein NEDDylation upon Pevonedistat treatment, hence the resistance. These results also provided the proof-of principle that NAE is indeed the target of these inhibitors through which induce their cytotoxic effects. In the current study though, Pevonedistat still reduces NEDDylation suggesting that the resistance has nothing to do with the NEDD8 pathway per se. Sequencing may be informative. The increase in the levels of NEDD8 is rather modest-at best-so I feel the claim that the effects in IFN γ and GzmB levels (Fig. 5G,H) are due to increase in NEDD8 are not fully supported. So, overall, I do not feel this is an appropriate system to assess the role of NEDD8 increase in ICB response. The authors may consider the knockout of the NEDP1 deneddylating enzyme that results in a dramatic increase in protein neddylation and it is not an essential gene, as a more appropriate approach.

Authors' response: We thank the reviewer to bring this knowledge to our attention. We agree with the reviewer that the increase in protein neddylation in resistant cells is modest. Therefore, we have followed the suggestion from the reviewer and generated a new CRISPR KO cell line, where *NEDP1* was deleted in MDA-MB-231 cells using CRISPR/Cas9 (**Figure S7A**). As shown in **Figure S7B**, the KO cells demonstrated strongly enhanced protein neddylation and retained neddylation of several previously unseen substrates or enzymes. However, *NEDP1* KO cells did not diminish immune activation in TICS (**Figure S7C**), which was a distinct phenotype as compared to pevonedistat-resistant cells. In contrast to the earlier paper, where pevonedistat lost its inhibitory function on protein neddylation due to mutations on *NAE β* (Milhollen, et al, Cancer Cell, 2012), protein neddylation in our resistant cells remained sensitive to drug treatment. Therefore, we think that treatment-driven mutations on the pathway are unlikely to contribute to the resistance. Given that pevonedistat engages with targets outside protein neddylation, e.g. ubiquitination, the lack of immunogenicity in pevonedistat resistant cells may not be a direct result of enhanced protein neddylation. In light of this new evidence, we have revised our conclusion and discussion regarding pevonedistat-resistant cells in line 245-249.

11. It is advised that for the presentation of the statistical analysis on the graphs, each replicate experiment is shown as different colour or symbol to be directly compared with the relevant replicate in different used conditions.

Authors' response: We thank the reviewer for this point. We have revised the graphs where possible, for example in **Figure 1A** and **Figure 4G**.

Reviewer #3 (Remarks to the Author):

1. Here the authors present 2 genome-wide CRISPR screens in their co-culture system with triple negative breast cancer cells. They found that MDA-MB-231 cells can compensate for NEDD8 loss and these deficient cells are vulnerable to nivolumab, presumably due to enhanced protein expression for antigen presentation (HLA-DRA, -DRB and CD74). Overall the paper is well organized and provides some original findings.

Authors' response: We thank the reviewer for this positive view on our work.

Major conceptual comment:

2. Targeting genes with CRISPR KO to sensitize immunotherapy is not new. The context of many previous studies should be carefully incorporated. The study is far from clinical impact because there's no specific inhibitors for NEDD8 and toxicities are unclear.

Authors' response: This point is well-noted. It is true that there has been precedence on using CRISPR KO or loss-of-function screens to identify immunotherapy resistance genes in cancer cells. However, none of the studies utilized genome-wide CRISPR/Cas9 screens to reveal resistance to clinically

approved PD-1/L1 blocking antibodies with a focus on human TNBC. We have now revised the manuscript to accommodate these changes in line 76-79 and line 346-351.

Because *NEDD8* was a strong hit from our genome-wide CRISPR screens, we hypothesized that pharmacological inhibitors against protein neddylation, i.e. pevonedistat, could bring benefits to patients by boosting the effects of immunotherapy. However, we observed that such compounds demonstrated a direct negative effects on adaptive immune cells, which probably due to functions outside protein neddylation. Although there are no current inhibitors against NEDD8, optimizing the dose and treatment intervals of current neddylation inhibitors may preserve the immune-activating functions. We certainly hope that this work would pave the way for clinical studies that could bring patient benefits in a long-run. We have expanded our discussion on this point in line 411-416.

The authors need to address the following points:

3. The finding that the MDA-MB-231 cancer cells can compensate for loss of an essential gene, NEDD8, is an interesting finding, but also raises the question if normal mammary cells can also compensate or if this is a unique finding of the case of triple negative breast cancer?

Authors' response: We thank the reviewer for bringing up this important point. To assess the essentiality of NEDD8 in non-malignant mammary cells, we utilized the MCF10A cell line, which is a well-studied non-tumorigenic mammary cell line. When the *NEDD8* gene was targeted by CRISPR/Cas9, MCF10A cells demonstrated substantial loss in proliferative capacity. In contrast to the tumorigenic TNBC cell lines, i.e. MDA-MB-231, HCC1937 and BT549, we were unsuccessful in generating a NEDD8-deficient MCF10A cell line under the 5-month period of this revision work (**Figures to reviewers R1**). Therefore, we hypothesize that compensation for the loss of essential genes is more likely in aggressively growing TNBC cells.

4. The authors show that NEDD8 KO MDA-MB-231 cells show depleted expression of genes involved in DNA repair and telomere maintenance. Have the authors evaluated DNA damage in these KO cells and their sensitivity to cytotoxic chemotherapy or radiation? This could also increase the clinical impact of their paper.

Authors' response: We thank the reviewer for this important note. Following the recommendation, we have tested the sensitivity of chemotherapeutic drugs, including paclitaxel, doxorubicin and fludarabin, on the NEDD8 competent and deficient cells. As shown in **Figure S4A**, we did not observe altered potency of these drugs on NEDD8 KO cells. Neither control nor NEDD8 KO cells were sensitive to fludarabin *in vitro*. Unfortunately, we currently do not have access to a radiation facility, and therefore was not able to assess the effects of radiation on NEDD8 KO cells. This has been commented in line 159-161.

5. For the following results: “Moreover, we observed that gRNAs targeting the NAE1 gene were depleted in both the WT and KO cells in the genome-wide screens (Figure 2C). NAE1 144 encodes the NEDD8-activating enzyme E1 subunit 1 (NAE1), which is a key subunit of the 145 first heterodimer enzyme of the neddylation pathway [16]. Pharmacological inhibitors against 146 NAE, i.e. pevonedistat [25] and TAS4464 [26], have been developed and tested in patients as 147 potential anti-cancer therapies [25, 27]. Cell proliferation assays showed that the WT and KO 148 cells were equally sensitive to pevonedistat (Figure 2E) and TAS4464 (Figure S3D). Deletion 149 of NEDD8 did not influence the expression of NAE1 protein (Figure S3E).”

Isn't this an unexpected finding given that the KO cells can survive without NEDD8 and NAE1 functions in this pathway. Does NAE1 have other functions besides NEDD8 activation which are essential for cells? This would be crucial to explain for the readers.

Authors response: We thank the reviewer for identifying this question. Indeed, it was initially surprising to us that pevonedistat could inhibit proliferation of NEDD8 deficient cells to the same degree as the parental cells. To examine the detailed mode-of-action for pevonedistat, we have now performed further experiments and showed that it could inhibit the proliferation of *NAE1* KO cells, which is the putative target of the compound (**Figure 5F and G**). Our additional data demonstrated that pevonedistat could inhibit ubiquitination at high concentrations (**Figure 5H-J**). These data support that the specificity or dose scheduling of current pharmacological inhibitors against protein neddylation should be optimized in combination with immunotherapy. We have added a new section in results to mention these data, in line 250-272.

6. For the following results: “To test whether NEDD8 KO TNBC cells can induce stronger immune cell activation, WT or 194 NEDD8 deficient MDA-MB-231 cells were co-cultured with primary human lymphocytes in 195 TICS +/- ICB drugs. The KO cells induced only marginal enhancements of soluble IFN γ and granzyme B in culture supernatants, as compared to the WT cells (Figure 4C). In contrast, we observed significantly increased production of these immune-activating cytokines in the 198 presence of nivolumab or durvalumab (Figure 4C; Figure S4B). In accordance, absence of NEDD8 in MDA-MB-231 cells induced stronger proliferation of CD8+ and CD4+ T cells in 200 TICS in response to blockade of either PD-1 or PD-L1 (Figure 4D). The activation of NK cells by NEDD8 KO TNBC cancer cells showed only a trend of increase (Figure S4C). The increased release of soluble granzyme B in response to nivolumab or durvalumab was further confirmed using an additional WT/KO cell line pair derived from HCC1937 cells (Figure 4E).”

Do the authors know why they did not see soluble IFN γ and granzyme B in co-culture supernatants of KO cells in the absence of PD1 blockade? Have they profiled their T cells for exhaustion markers that would explain this?

Authors response: We thank the reviewer for pointing out this. Soluble factors, e.g. IFN γ and Granzyme B, were detectable in the co-culture in **Figure 4C** and **4E**, and were dependent on the ratio between lymphocytes and cancer cells. Moreover, we observed a trend of increase when comparing KO to parental cells (grey vs white bars) but this difference did not reach statistical significance. When ICB drugs were added, the production of these cytokines were strongly elevated, leading to statistically significant differences in co-cultures with KO cells. We have improved the results session linked to this figure to provide better clarity in line 205-206.

We agree with the reviewer that it is important to assess T cell status at the end of TICS and have performed additional analysis to characterize expression of surface markers on T cells and displayed the results in **Figures to reviewers R2**. We observed strong upregulation of activation marker CD25 on proliferating CD8+ T cells activated by TNBC cells, but these cells lack exhaustion markers such as PD1, TIGIT and TIM3 and express LAG3 at low levels. Moreover, CD127 was not clearly down-regulated in proliferating CD8+ T cells from TICS. According to literature (e.g. Belk et al., Cancer Cell, 2022), induction of T cell exhaustion *in vitro* requires repeated and chronic TCR stimulation. In TICS, naive lymphocytes were exposed to allogeneic antigens on cancer cells for 5 days without repeated challenge and therefore may not be sufficient to induce exhaustion.

7. The authors have done excellent mouse studies showing the NEDD8 delayed growth of E0771 tumors in B6 mice only upon PD1 blockade. Did they analyze tumor infiltrating lymphocytes in NEDD8 KO tumors and WT tumors with and without PD1 blockade and see increased proliferation and release of soluble IFN γ and granzyme B?

Authors' response: We appreciate the recognition of the reviewer on our mouse studies and thank the reviewer for this important comment. We have now performed additional *in vivo* experiments and added

this data in a new figure, **Figure 7**. In these studies, we analyzed the infiltrating immune cells using flow cytometry as well as a nanostring panel that measures 800 immune-related transcripts. Frequency of tumor-infiltrating T cells were comparable among groups but KO tumors receiving PD1 blockade showed enhanced expression of CD25 and TNF α . When mapping the intratumoral immune landscape using nanostring, we observed profound enhancement of IFN responsive genes, e.g. *Mx1*, *Lsg15*, *Ifit1*, *Cxcl10* and *Ifitm1* and immune activating genes, e.g. *Il2ra/cd25*, *Cd80* and *Gzma* in KO mice treated with anti-PD1. Overall, we concluded that PD1 blockade elicits a stronger pro-inflammatory response in NEDD8 deficient tumors. This section is now in line 300-330.

8. The impact and significance of their finding could be increased if the authors analyzed publicly available immunogenomic patient datasets to see if increased expression of NEDD8, or the neddylation pathway in general, is predictive of poorer response to immunotherapy. Could this be used as a biomarker?

Authors' response: We fully agree with the reviewer that publicly available patient datasets could add significance to our work. Although such datasets are accessible for certain cancers such as melanoma or bladder cancer, datasets generated from ICB-treated TNBC patients are scarce. We obtained and analyzed a dataset of a small patient cohort, who were treated with or without an induction therapy, followed by nivolumab (Voorwerk et al., Nature Medicine, 2019). The goal of the TONIC trial was to study whether neoadjuvant therapies in combination with nivolumab could increase the response rate and survival.

To be able to investigate the predictive value of a certain gene, it needs to demonstrate variable expression among donors to allow the establishment of a threshold between high and low samples. Expression of *NEDD8* mRNA in pre-therapy samples showed little variations among donors (**Figures to reviewers R3A**). Next, we calculated the average expression and the standard deviation of each gene across all pre-therapy samples (**Figures to reviewers R3B**). *NEDD8* mRNA was highly expressed among samples but demonstrated very little variation. Moreover, we observed the same expression pattern of *NEDD8* mRNA in samples collected after induction therapy but before nivolumab (**Figures to reviewers R3C and D**). This is not unexpected because NEDD8 and protein neddylation are essential for cell homeostasis and are expressed by a wide range of cell types in tumor tissues. Based on this analysis, we concluded that *NEDD8* mRNA cannot be used as a predictive biomarker in this patient cohort due to its high and indifferent expression among patients.

Due to an existing Data Transfer Agreement (DTA), we have decided not to include this initial analysis in the main figures of this work.

Minor points:

1. In paragraph 2 of the results section please mention how long the cells were co-cultured. I also could not find this information in their methods section. This would allow people to reproduce their results.

Authors' response: We are sorry that this information was not more evident. The culture time was mentioned in the illustration in Figure 1B. We have now added this information also in the text in line 99.

2. Figure 6D- Please show WT cells in the same plot under the same conditions

Authors' response: We thank the reviewer for this point. EO771 WT tumors were not responsive to PD-1 blockade under this experimental condition (**Figure 6C**) so we focused on the effects of CD8 T cell depletion in the KO tumors. This has been clarified in line 296-298.

3. Sup Figure 1E- It is hard to see a difference in media color between conditions. Were total cell numbers counted?

Authors' response: Thank you for this good point. We have now added the alive cancer cell number at the end of the CRISPR screen to **Figure S1E**.

4. Sup Figure 4B- Images are too small to see.

Authors' response: We thank the reviewer. Figure S4B, now as **Figure S5A**, has been enlarged in the revised manuscript.

REVIEWER COMMENTS

Reviewer #1 (Remarks to the Author):

All the concerns have been addressed. Thus, I recommend this manuscript for publication.

Reviewer #2 (Remarks to the Author):

The manuscript has been improved with the addition of several new experiments. In my initial review, I felt that at least one of the 2 main aspects of the manuscript had to be strengthened, leaving it up to the authors to decide.

With regards to the NEDD8 KO phenotype, I think it is a rather intriguing finding. I can assure the authors that for this finding to have a significant impact in the field, assessing key aspects of neddylation and in particular the activity of cullin-ring-ligases is critical. It is advised to follow the strategy used in the original Pevonedistat paper by Soucy et al. 2009, PMID: 19360080.

The authors have performed some experiments towards this but there are still some issues:

Regarding the activity of nedd8 enzymes:

Monitoring the thioester charging of the E1/E2 enzymes. For this the authors need to prepare extracts and analyse them in SDS-PAGE with no reducing agents and preferably low pH (2-3) to preserve the thioester bond. There are several protocols for this. Currently based on the provided info, the analysis was performed under reducing conditions and thus the modification observed in experiments including Fig. 5 is a covalent lysine modification and not a thioester bond on the catalytic cysteine.

Figure 5

J: Is Pevonedistat applied in parental MDA-MB-231 cells or the NEDD8 KO? If it's the parental cells how do we know it is Ub-NAE1?

I: Why PYR41 would decrease NAE1 modification? Does this imply this is ubiquitination of

NAE1? rather than monitoring thioester formation of NAE1 possibly with Ub? the decrease could be due to non-specific effect of PYR41 at high doses?

Advised to use the high specific UBA1 inhibitor TAK-243 (MLN7243) that blocks UBA1 at nM doses.

Figure S4

The migration should be improved so the modified and unmodified cullin should be clearly resolved. Treatment with Pevonedistat and TAK-243 should indicate the identity of the modified form of cullin. If for some reasons, low levels of cullin neddylation (and subsequently cullin-ring-ligase activity) is still preserved in NEDD8 KO cells, low doses (100-200nM) of Pevonedistat should eliminate this band. Similarly, if this is ubiquitinated cullin, low doses of TAK-243 (500nM or so) should eliminate this band.

Based on the original Soucy et al. studies the cytotoxic effects (re-replication) of Pevonedistat are mainly due to stabilization of p21 and cdt1, which are controlled by cullin 1 and cullin 4A/B. It is strongly advised to at least test the modification of these cullins and p21/cdt1 levels in the absence/presence of Pevonedistat/TAK-243 in control and NEDD8 KO cells.

The fact that Pevonedistat reduces at high doses total ubiquitination is expected (also shown in Soucy et al.) due to inhibition of cullin-ring-ligases. This should be considered in the text.

The following references should be included:

G Wei Xu et al. 2014, PMID: 24691136 with the Milhollen et al. 2012

Leidecker et al., 2012, PMID: 22370482 with the Hjerpe et al. 2012

Reviewer #3 (Remarks to the Author):

The authors addressed some of the points, and provided several piece of new data, however, concerns remain.

While the reviewer appreciate certain experimental attempts and textual arguments, the

revised work did not further support NEDD8 as an important target. It is interesting that its KO in cancer cell led to increased sensitivity, pharmacological inhibition showed exactly the opposite. The unknown possible immune effect is vague and there's no data on immune cell targeting with NEDD8.

The human translation and clinical relevance need to be strengthened. Is there prognostic advantage by targeting NEDD8? Any clinical benefit?

In addition, the authors can not generate cell lines and didn't test the general applicability thus the implication may be limited.

The DNA damage signal is interesting, however it was not followed up. Understand radiation equipment is not available everywhere, but many DNA damaging agents are available to mimic.

Point-by-point response letter to reviewers

Reviewer #2

1. The manuscript has been improved with the addition of several new experiments. In my initial review, I felt that at least one of the 2 main aspects of the manuscript had to be strengthened, leaving it up to the authors to decide.

Authors' reply: We thank your input on the mechanisms of protein neddylation. We have further developed the results regarding protein neddylation and ubiquitination in a series of new experiments below. In particular, it was intriguing that the ubiquitination inhibitor of your choice, i.e. TAK-243, induced a much stronger decrease in the proliferation of *NEDD8* KO cells, as compared to the control cells (**Figure 1**). Although protein neddylation could be conducted through the ubiquitin system as a result of an increased NEDD8/ubiquitin ratio, it remains unclear whether the ubiquitin system could compensate for the loss of NEDD8. Considering that ubiquitin enzymes become strongly expressed and more functionally relevant in KO cells, we propose that the absence of NEDD8 promotes a functional switch in cancer cells so protein ubiquitination becomes dominant and indispensable. Moreover, we would like to highlight that silencing the *NEDD8* gene by CRISPR/Cas9 results in a sharp initial decrease in cell proliferation, which requires a long culture period to overcome. Therefore, phenotype and protein regulatory network in the KO cells may be different to acute responses induced by neddylation inhibitors in a short treatment period. We have now removed the cell proliferation data with PYR41 and added the new data with TAK-243 in Figure 3E. We have updated the illustration in Figure 2H to highlight the importance of protein ubiquitination in KO cells and also discussed the implication of our new results in lines 170-171, 182-199, 380-388.

Figure 1. Effects of TAK-243 on control or *NEDD8* KO MDA-MB-231 cells. Control or *NEDD8* KO MDA-MB-231 cells were seeded in presence of the ubiquitination inhibitor, TAK-243. Cell proliferation was monitored using live-cell imaging in an Incucyte instrument.

2. With regards to the *NEDD8* KO phenotype, I think it is a rather intriguing finding. I can assure the authors that for this finding to have a significant impact in the field, assessing key aspects of neddylation and in particular the activity of cullin-ring-ligases is critical. It is advised to follow the strategy used in the original Pevonedistat paper by Soucy et al. 2009, PMID: 19360080.

Authors' reply: We agree with you that the original paper by Soucy and colleagues provided an excellent example on assessing the inhibition of protein neddylation by pevonedistat and we have followed your advice and investigated different aspects of pevonedistat in our experimental system. Overall, our results are in agreement with literature data, where pevonedistat potently inhibits protein neddylation and constrains cancer cell proliferation. However, the neddylation-independent effects of pevonedistat have been demonstrated previously. For example, pevonedistat could trigger cell death through the modulation of c-FLIP levels (PMID: 21914854). It has also demonstrated effects on oncogenic pathways including EGFR, MYC or RAS/MEK (PMID: 27162365). Moreover, pevonedistat could alter mitochondrial fitness and energy consumption in breast cancer cells (PMID: 30668548). In this manuscript, our work revealed the possible neddylation-independent function of pevonedistat on cancer cell proliferation and immune cell activation through the inhibition of ubiquitination at high concentrations. These data do not de-validate pevonedistat as a neddylation inhibitor, but offer new perspectives for optimized clinical utility in combination with immunotherapy. We have expanded our discussions in lines 413-416, 437-447.

We agree with you that it would be fascinating to dissect the effects of pevonedistat on *NEDD8* deficient cells. However, this will require major investments in my lab to establish new research expertise and instrumentation in protein science, biochemistry, chemoproteomics, which is unfortunately not feasible at the moment. We are committed to keep pursue this line of research and actively seek collaborators who can develop the concept in a follow-up project.

The authors have performed some experiments towards this but there are still some issues:

Regarding the activity of nedd8 enzymes:

3. Monitoring the thioester charging of the E1/E2 enzymes. For this the authors need to prepare extracts and analyse them in SDS-PAGE with no reducing agents and preferably low pH (2-3) to preserve the thioester bond. There are several protocols for this. Currently based on the provided info, the analysis was performed under reducing conditions and thus the modification observed in experiments including Fig. 5 is a covalent lysine modification and not a thioester bond on the catalytic cysteine.

Authors' reply: We thank you for sharing your expertise on the thioester charging of E1/E2 enzymes. We have carefully reviewed our protocols and confirmed that no reducing reagents were included during the preparation of cell lysates. This has been specified in line 674-675.

We agree that it is important to investigate whether thioester bonds of modified NAE1 enzyme are preserved in our assays. Therefore, we have performed additional experiments using reducing reagents dithiothreitol (DTT) or beta-mercaptoethanol (BME) during sample preparation, according to a paper you suggested (Hjerpe, et al., 2012). As you correctly anticipated, the presence of reducing reagents during sample preparation abolished the modified NAE1 in western blots, demonstrating that thioester bonds were preserved in our assays.

Figure R1. Effects of reducing agents on the detection of modified NAE1 in western blotting. Reducing agents beta-mercaptoethanol (BME, 5%) or dithiothreitol (DTT, 100 mM) were added during the extraction of cell lysates from control MDA-MB-231 cells. Expression of NAE1 protein was determined using western blotting.

Figure 5

4. J: Is Pevonedistat applied in parental MDA-MB-231 cells or the NEDD8 KO? If it's the parental cells how do we know it is Ub-NAE1?

I: Why PYR41 would decrease NAE1 modification? Does this imply this is ubiquitination of NAE1? rather than monitoring thioester formation of NAE1 possibly with Ub? the decrease could be due to non-specific effect of PYR41 at high doses?

Advised to use the high specific UBA1 inhibitor TAK-243 (MLN7243) that blocks UBA1 at nM doses.

Authors' reply: These questions are very much appreciated. You are correct that we treated the MDA-MB-231 parental cells with pevonedistat and measured the expression of NAE1 by western blot. We agree with you that high concentrations of PYR41 in this assay may result in unspecific effects. Therefore, we have followed your suggestion and repeated the experiment by treating MDA-MB-231 control cells with TAK-243. As shown in **Figure 2**, TAK-243 inhibited NAE1 modification between 100-400 nM, suggesting that UBA1 regulates the modification of NAE1 enzyme, possibly through ubiquitination.

Figure 2. Effects of TAK-243 on the modification of NAE1 in MDA-MB-231 cells. Parental MDA-MB-231 cells were treated with TAK-243 at different concentrations in a 6 well plate. DMSO was used as a control. Cells were harvested after 24 hours and levels of NAE1 was measured using western blotting.

NAE enzyme is believed to specifically activate NEDD8. However, there has been examples where ubiquitination was detected after immunoprecipitation of FLAG-NAE1 in the anti-FLAG agarose (PMID: 35582972). Because the modified NAE1 in our assays preserves thioester bonds, it is possible that NAE1 could conjugate to ubiquitin through these bonds, as

you have suggested. Moreover, it was reported that Ub and N8 can form substrate free heterodimers (PMID: 23105008) and NAE was able to activate several ubiquitin (Ub) variants at a similar level as NEDD8 (PMID: 23936405, 14690597). If such heterodimers or Ub variants are present in the TNBC cell lines, it is possible that NAE1 is conjugated with a mixture of Ub and N8. These intriguing hypotheses could be mechanistically investigated by experts in the field but are beyond the scientific expertise of our laboratory. We have added this data in Figure 5I and moved the results with PYR41 to Figure S7D. This is commented in lines 280-284.

5. Figure S4

The migration should be improved so the modified and unmodified cullin should be clearly resolved. Treatment with Pevonedistat and TAK-243 should indicate the identity of the modified form of cullin. If for some reasons, low levels of cullin neddylation (and subsequently cullin-ring-ligase activity) is still preserved in NEDD8 KO cells, low doses (100-200nM) of Pevonedistat should eliminate this band. Similarly, if this is ubiquitinated cullin, low doses of TAK-243 (500nM or so) should eliminate this band.

Authors' reply: We thank you for this important point and agree that the separation between modified and non-modified cullin-1 should be improved. In order to resolve this issue, we have performed new analysis using western blotting and optimized the running time. We also thank you for the advice on compound treatments and have performed these experiments. In MDA-MB-231 control cells, pevonedistat abrogated cullin-1 modification at low concentrations, while TAK-243 showed limited effects (**Figure 3**). In *NEDD8* KO cells, TAK-243 showed a notable inhibition of cullin-1 modification at concentrations above 100 nM. This inhibition by pevonedistat in KO cells was clear but less complete at low concentrations, e.g. 10 nM (**Figure 3**). Although cullin-1 modification may be less important for the proliferative capacity of MDA-MB-231, i.e. TAK-243 decreased cell proliferation without substantially impacting cullin-1 modification, these data could indicate the increased involvement of the ubiquitination system upon NEDD8 deletion. These new results have been added to Figure S4C and mentioned in lines 204-205.

Figure 3. Effects of pevonedistat or TAK-243 on the modification of cullin-1 in MDA-MB-231 cells. Parental or NEDD8 KO MDA-MB-231 cells were treated with pevonedistat or TAK-243 at different concentrations in a 6 well plate. DMSO was used as a control. Cells were harvested after 24 hours and levels of NAE1 was measured using western blotting.

6. Based on the original Soucy et al. studies the cytotoxic effects (re-replication) of Pevonedistat are mainly due to stabilization of p21 and cdt1, which are controlled by cullin 1 and cullin 4A/B. It is strongly advised to at least test the modification of these cullins and p21/cdt1 levels in the absence/presence of Pevonedistat/TAK-243 in control and NEDD8 KO cells.

Authors' reply: Your comments are appreciated. We have now performed new experiments, where the expression of CDT1 was tested in control or NEDD8 KO MDA-MB-231 cells, treated with pevonedistat or TAK-243. We can observe in **Figure 4** that pevonedistat and TAK-243 induced the stabilization of CDT1 in MDA-MB-231 cells in a dose-dependent manner. NEDD8 deficiency did not lead to changes of CDT1 stabilization upon pevonedistat treatment, which correlates to the effects on cell proliferation (Figure 2E). However, we observed that TAK-243 induced stronger stabilization of CDT1 in KO cells between 100 and 400 nM, which was in line with the enhanced inhibition on cell proliferation in KO cells by this compound. These results show that both neddylation and ubiquitination regulate CDT1 degradation. We have added these new results as Figure 3F and 3G and mentioned the results in lines 200-204.

Figure 4. Pevonedistat and TAK-243 trigger stabilization of CDT1 in MDA-MB-231 cells. Control or NEDD8 KO MDA-MB-231 cells were treated with various concentrations of pevonedistat or TAK-243 in a 6-well plate. DMSO was used as controls. Cell pellets were collected after 24 hours and expression of CDT1 was determined in cell lysates using Western Blotting.

7. The fact that Pevonedistat reduces at high doses total ubiquitination is expected (also shown in Soucy et al.) due to inhibition of cullin-ring-ligases. This should be considered in the text.

Authors' reply: We thank you for this comment and agree that this addition will improve our manuscript. Please find this change in lines 412-413.

8. The following references should be included:

G Wei Xu et al. 2014, PMID: 24691136 with the Milhollen et al. 2012

Leidecker et al., 2012, PMID: 22370482 with the Hjerpe et al. 2012

Authors' reply: Thank you for suggesting these references. We have added them to the revised version of the manuscript together with the existing references.

Reviewer #3

1. The authors addressed some of the points, and provided several piece of new data, however, concerns remain. While the reviewer appreciate certain experimental attempts and textual arguments, the revised work did not further support NEDD8 as an important target.

Authors' reply: We thank you for taking the time to review our manuscript and we are sorry that the key message regarding pharmacological inhibitors against protein neddylation is unclear in our manuscript. According to genetic screening data, the *NEDD8* gene is important for the survival of more than 1000 human cancer cell lines (<https://depmap.org/portal/gene/NEDD8?tab=overview>). Therefore, targeting protein neddylation has been considered a promising approach to limit cancer growth.

To confirm the clinical relevance of NEDD8 as a target, we tested the expression of NEDD8 in paired tumor and adjacent normal tissues. We observe that NEDD8 protein is expressed at higher levels in melanoma and bladder tumor tissues, in comparison to adjacent normal tissues (Figure R2). This means that cancer cells could be more sensitive to protein neddylation inhibition. Despite our best efforts, we were not able to obtain paired samples from breast cancer patients. To strengthen the clinical relevance, we have analyzed the relationship between NEDD8 mRNA expression and response to chemo-immunotherapy in point 4 below.

Figure R2. Expression of NEDD8 protein is elevated in human tumor tissues. Sections of tumor and adjacent normal tissues were obtained from patients with bladder cancer or melanoma. Expression of NEDD8 protein was determined using immunohistochemistry.

2. It is interesting that its KO in cancer cell led to increased sensitivity, pharmacological inhibition showed exactly the opposite.

Authors' reply: We thank you for raising this point. As you correctly pointed out, KO cells demonstrated increased sensitivity to nivolumab in co-culture with immune cells, while the pharmacological inhibitor only showed this increase at medium-low concentrations (Figure 5C). Our detailed mechanistic investigations revealed that it was due to neddylation-independent effects of these compounds at high concentrations, e.g. inhibition of ubiquitination. This is an

important note to consider because identifying the immune-agonistic dose or treatment schedule could be key for the clinical success of these neddylation inhibitors. We have expanded our discussion on this point in the revised manuscript, in lines 413-415, 442-447.

3. The unknown possible immune effect is vague and there's no data on immune cell targeting with NEDD8.

Authors' reply: We thank you for mentioning this point. Our central hypothesis in the current study is that *NEDD8* deletion in cancer cells enhances response to PD-1/L1 blockade therapy. We show that *NEDD8* deficient cancer cells re-shape tumor micro-environment and are more responsive to ICB therapy.

Because NEDD8 is expressed by immune cells, we agree that it is of interest to investigate its function. Because available neddylation inhibitors do not directly target NEDD8 and demonstrate NEDD8-independent effects, it is more appropriate to employ CRISPR/Cas9-based methods to specifically disrupt *NEDD8* gene expression in healthy human immune cells. To explore whether silencing of the *NEDD8* gene can be achieved in primary human immune cells, we introduced the RNP complexes using our previously tested protocol (PMID: 36757800) in healthy monocytes. This procedure resulted in the deletion of free NEDD8 protein but only marginally reduced neddylated proteins (Figure R3). Considering the long cell culture time needed to achieve stable NEDD8 KO in cancer cells, significant technical optimization is required to allow the investigation of NEDD8 biology in primary human immune cells. The alternative is to create immune cell lineage specific *NEDD8* KO mouse models, which we do not currently have the financial means for. Recognizing that many studies in the literature apply neddylation inhibitors on immune cells, we have expanded our discussion on this aspect in the revised manuscript, in lines 436-440.

Figure R3. Genetic targeting of the NEDD8 gene in primary human monocytes. Fresh primary human monocytes were isolated from buffy coats using CD14 positive selection and RNP complexes with or without the NEDD8 targeting gRNA were introduced using electroporation. Cells were maintained in IMDM medium containing 100 ng/ml rhGM-CSF and 10% human AB serum, followed by analysis using western blotting.

4. The human translation and clinical relevance need to be strengthened. Is there prognostic advantage by targeting NEDD8? Any clinical benefit?

Authors' reply: We thank you for this comment. Because protein neddylation is required to sustain the uncontrolled growth of cancer cells, high expression of *NEDD8* mRNA or enzymes regulating protein neddylation has been associated with worse clinical outcome in several human cancer types (reviewed in PMID: 29331584). Our results in this manuscript elucidates the novel role of NEDD8 as a cancer intrinsic resistance mechanism against PD-1/L1 blockade therapy. Therefore, the predictive value of NEDD8 mRNA in breast cancer patients receiving ICB therapy should be explored. To address this question, we retrieved published sequencing results from breast cancer patients treated with paclitaxel or paclitaxel+pebrolizumab in the I-SPY2 neoadjuvant platform trial (PMID: 35623341). We observe that patients with low or medium *NEDD8* mRNA expression experienced more frequent pathologic complete responses (pCR) after the combination therapy, i.e. 45.5% and 51.8%, respectively, as compared to patients in the *NEDD8* high subgroup (22.2%). This analysis provides an early indication on NEDD8 as a potential predictive biomarker for ICB therapy in breast cancer patients. We have now included this data in Figure 1E and mentioned it in line 121-127. We have also discussed the implication and limitation of this data in line 427-432. The data analysis process is added to line 862-868.

5. In addition, the authors can not generate cell lines and didn't test the general applicability thus the implication may be limited.

Authors' reply: We thank you for this question and sorry for not communicating our results clearly. As expected, targeting the *NEDD8* gene by CRISPR/Cas9 caused significant loss of cell viability in human cancer cell lines. Tumorigenic breast cancer cell lines, i.e. MDA-MB-231, HCC1937, BT549 and EO771, were able to reprogram cellular process and adapt to the loss of NEDD8 protein.

In order to answer your previous question on whether normal breast cells could adapt to the loss of *NEDD8*, we performed the same CRISPR/Cas9 KO procedure using a MCF10A cell line, which was derived from human fibrocystic mammary tissue more than 30 years ago (PMID: 1975513). The cell line is non-tumorigenic and is widely used as a model to study regulation of normal breast epithelial phenotype. In contrast to the malignant and tumorigenic breast cancer cell lines, MCF10A failed to survive the loss of the *NEDD8* gene. This can be partially explained by the largely normal genome and cellular function in this cell line (PMID: 34400468), leading to weaker ability to adapt to genetic alteration and cellular stress. We have expanded our discussion to highlight this point in line 393-396.

6. The DNA damage signal is interesting, however it was not followed up. Understand radiation equipment is not available everywhere, but many DNA damaging agents are available to mimic.

Authors' reply: We thank you for the good question. Using proteomics and genome-wide CRISPR screens, we observed that KO cells demonstrated enhanced cell cycle and DNA replication potential, in order to sustain cell proliferation in the absence of NEDD8. Moreover, we observed a dependency of 'transcription-coupled nucleotide-excision repair (TC-NER)' in KO cells, which is a specific mechanism to repair damage on the transcribed strand of an active gene (reviewed in PMID: 11823795). A previous study demonstrated that multiple DNA repair mechanisms, in addition to TC-NER, are required to confer resistance by removing damages caused by platinum-based chemotherapy drugs (PMID: 30137419). It is possible that changes of TC-NER alone in KO cells is not sufficient to alter sensitivity to chemotherapy or other agents. For clarity, we have replaced 'DNA repair' in Figure 2H with 'DNA replication'. We have also checked throughout the text to avoid any possible confusions.

REVIEWERS' COMMENTS

Reviewer #2 (Remarks to the Author):

Comments:

For the following response:

"However, the neddylation-independent effects of pevonedistat have been demonstrated previously. For example, pevonedistat could trigger cell death through the modulation of c-FLIP levels (PMID: 21914854). It has also demonstrated effects on oncogenic pathways including EGFR, MYC or RAS/MEK (PMID: 27162365). Moreover, pevonedistat could alter mitochondrial fitness and energy consumption in breast cancer cells (PMID: 30668548)".

I am not sure where this conclusion of neddylation-independent effects of pevonedistat is based on. Indeed, the PMID: 21914854 indicates this, but is based only on data derived from nedd8 knockdown (not ideal as CRL function is still preserved under these conditions). The other mentioned studies do not indicate this. I think the most accurate description of the data is CRL-independent functions of NEDD8 that includes neddylation of non-cullin substrates, not necessarily implicated in protein degradation (many such examples exist in the literature). This can also explain why the nedd8 KO cells are more sensitive to TAK-243 (degradation independent functions of nedd8).

The data presented in the new figures 3 and 4, at least to me, provide quite convincing evidence that in nedd8 KO cells, cullin modification and CRL activity is preserved. The fact that cullin modification is more sensitive to Pevonedistat compared to TAK-243, strongly indicates that somehow the nedd8 KO cells preserve cullin neddylation?

I feel that the CRL independent aspect of neddylation is a rather important point to be extensively discussed.

Reviewer #3 (Remarks to the Author):

The authors provided additional arguments and speculation on NEDD8 loss vs neddylation /

ubiquitination. However, the mechanism is still unclear. To support the title / main claim that "NEDD8 loss causes cancer vulnerability to immune checkpoint blockade in triple-negative breast cancer", biochemical experiments and evidence need to be provided. Without mechanistic dissection, it is inaccurate to conclude that this is the effect from Nedd8 loss, as state in the title.

However, if the biochemical experiments are not feasible at all from the authors' labs/sites, the title and interpretation need to be changed. The data need to be interpret as is. The unclear / open parts without data support need to be revised and discussed clearly.

Point-by-point response letter to reviewers (NCOMMS-23-12581C)

Reviewer #2

1. "However, the neddylation-independent effects of pevonedistat have been demonstrated previously. For example, pevonedistat could trigger cell death through the modulation of c-FLIP levels (PMID: 21914854). It has also demonstrated effects on oncogenic pathways including EGFR, MYC or RAS/MEK (PMID: 27162365). Moreover, pevonedistat could alter mitochondrial fitness and energy consumption in breast cancer cells (PMID: 30668548)".

I am not sure where this conclusion of neddylation-independent effects of pevonedistat is based on. Indeed, the PMID: 21914854 indicates this, but is based only on data derived from nedd8 knockdown (not ideal as CRL function is still preserved under these conditions). The other mentioned studies do not indicate this. I think the most accurate description of the data is CRL-independent functions of NEDD8 that includes neddylation of non-cullin substrates, not necessarily implicated in protein degradation (many such examples exist in the literature). This can also explain why the nedd8 KO cells are more sensitive to TAK-243 (degradation independent functions of nedd8).

Authors' response: We thank you for this comment and agree that the discussion on this point can be improved. We have now removed the referred sentences and expanded discussion on the CRL-independent mechanisms of neddylation (please see below).

2. The data presented in the new figures 3 and 4, at least to me, provide quite convincing evidence that in nedd8 KO cells, cullin modification and CRL activity is preserved. The fact that cullin modification is more sensitive to Pevonedistat compared to TAK-243, strongly indicates that somehow the nedd8 KO cells preserve cullin neddylation? I feel that the CRL independent aspect of neddylation is a rather important point to be extensively discussed.

Authors' response: This comment is appreciated. We agree that CRL-independent neddylation is an important point to include in the discussion, in lines 412-418. We have added a comment regarding CRL modification in the discussion in lines 410-411.

Reviewer #3

The authors provided additional arguments and speculation on NEDD8 loss vs neddylation / ubiquitination. However, the mechanism is still unclear. To support the title / main claim that "NEDD8 loss causes cancer vulnerability to immune checkpoint blockade in triple-negative breast cancer", biochemical experiments and evidence need to be provided. Without mechanistic dissection, it is inaccurate to conclude that this is the effect from Nedd8 loss, as stated in the title.

However, if the biochemical experiments are not feasible at all from the authors' labs/sites, the title and interpretation need to be changed. The data need to be interpreted as is. The unclear / open parts without data support need to be revised and discussed clearly.

Authors' response: We thank you for this comment. Although NEDD8 KO cells showed a more immunogenic phenotype, i.e. enhanced HLA-DR expression, we can agree that the precise mechanisms linking NEDD8 loss to enhanced efficacy of PD-1 blockade remain to be explored. We have discussed this limitation in the last paragraph of the manuscript.